# WEIGHT SHARING FOR GRAPH STRUCTURED DATA

## ABSTRACT

Weight sharing is a key principle of machine learning. While well established for regular domains such as images, extending weight sharing to graphs remains challenging due to their inherent irregularity. We address this gap with a novel weight-sharing paradigm that indexes weights directly by graph invariants, i.e., functions preserved under node permutations. This formulation enables systematic reuse of parameters across structurally equivalent subgraphs, providing a principled mechanism for permutation-aware learning. To demonstrate the practicality of the approach, we introduce ShareGNNs, a new family of permutation-invariant graph neural networks that instantiate invariant-based weight sharing in a simple encoder-decoder design. We prove that the expressivity of ShareGNNs is lower-bounded by the discriminative power of the chosen invariant, allowing dynamic control of complexity. Experiments on subgraph counting, synthetic, and real-world benchmarks show that ShareGNNs achieve competitive performance on graph-level classification and regression tasks while using only one message-passing layer. Moreover, we discuss how the approach enhances interpretability and transferability.

## 1 INTRODUCTION

Weight sharing for graph-structured data is challenging because graphs are inherently irregular. In convolutional neural networks (Rumelhart et al., 1986; LeCun et al., 1989), a single kernel is reused across spatial locations, enabling models to capture local patterns efficiently and generalize with far fewer parameters. Extending this principle to graphs is not straightforward: unlike images, graphs lack a canonical node ordering, and their connectivity can vary greatly across instances. Yet, the ability to detect local patterns, such as rings in molecules or cliques in social graphs, is as essential for graphs as detecting edges or textures is for images. This motivates our central question:

*How can weight sharing be defined for irregular structures such as graphs?*

In this work, we address this challenge by introducing a new paradigm for weight sharing based on graph invariants, i.e., functions that remain unchanged under node permutations. We call this paradigm *invariant-based weight sharing*. By indexing weights directly with invariant properties (e.g., atomic numbers or node degrees), we obtain a dynamic weight-sharing scheme that applies to any node or node pair with the same invariant signature, independent of graph size or ordering. Beyond graphs, the paradigm naturally extends to other domains with pairwise relations, such as images (neighborhoods of pixels) or text (relations between words), see Appendix F.

Demonstrating the strength of our paradigm, we consider learning on graphs. This yields ShareGNN, a new family of message-passing graph neural networks that: (i) use invariant-based weight sharing, (ii) enable long-range interactions in a single message-passing layer, and (iii) offer principled control of expressivity through the choice of invariants. Conceptually, ShareGNN can be viewed as a message-passing framework with graph transformer-like connectivity: Like graph transformers (Yun et al., 2019; Shi et al., 2021), it allows all node pairs to interact within a single layer. The key difference is that these interactions are indexed by structural invariants rather than learned attention weights. Classical message-passing networks (Gilmer et al., 2017; Kipf & Welling, 2017) iteratively update node embeddings by aggregating information from neighbors, while graph transformers extend this idea by allowing every pair of nodes to exchange information via learned attention. Although both approaches are powerful, their parameters are tied to specific edges or attention scores measuring relations based on the current node feature representation. Thus, these approaches prevent natural

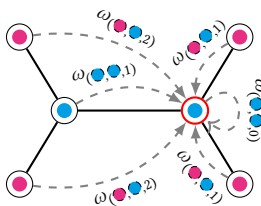 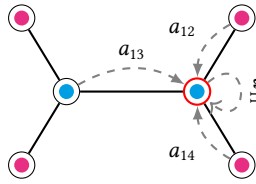

Figure 1: Invariant-based message passing (left) and graph attention (right) for the molecular graph of ethylene (hydrogen, carbon). The arrows denote the shared weights (left) and attention coefficients (right) for some target marked with red border.

weight sharing across structurally similar regions within or across graphs. Consequently, message-passing GNNs often require many layers and suffer from over-smoothing, while graph transformers, despite capturing long-range dependencies in a single layer, lack principled mechanisms for weight sharing and interpretability. ShareGNN takes a different path: using graph invariants to index weights, it organizes parameter sharing around structural patterns. This hybrid design combines the global connectivity of transformers with the inductive bias of message passing, resulting in models that are interpretable, transferable, and empirically strong.

The following toy example on molecular graphs illustrates our idea. Two node pairs are considered equivalent if: (i) their atomic numbers and (ii) their distances match. In Figure 1 (left) there are four node pairs (●, ●) that are two hops apart. All such pairs share a single weight, indexed by the tuple (●, ●, 2). In this way, every node pair in any molecule can be assigned a weight based on its atomic numbers and pairwise distance.

In ShareGNNs, this concept is generalized: messages are computed for every source-target pair of nodes. For a given pair, the source node's embedding is multiplied by the weight associated with that pair's invariant signature, and the result is added to the target node's embedding. Since these weights are shared using graph invariants, the same parameter is reused for all equivalent pairs, both within a graph and across different graphs. This mechanism enables information to flow between arbitrary nodes in a single layer, rather than being restricted to immediate neighbors, and makes the learned weights naturally transferable to new graphs that contain similar structural patterns. Again, we emphasize that graph transformers and ordinary message passing networks infer weights from learned node features rather than from structural invariants. Figure 1 (left) illustrates this process: invariant-based shared weights allow non-adjacent node pairs to send messages based on structural similarity. In contrast, GAT (Velickovic et al., 2017) (right) assigns distinct attention coefficients, based on node features, to each edge and aggregates information only from direct neighbors.

Having sketched our key ideas, we conclude with the main contributions of this work.

1) *Invariant-Based Weight Sharing:* A novel weight sharing paradigm using graph invariants, enabling systematic reuse of weights across structurally equivalent nodes and node pairs.

2) *ShareGNN:* A novel encoder–decoder architecture using invariant-based weight sharing that combines message passing with transformer-like connectivity.

3) *Theoretical and Empirical evaluation:* We prove that the expressivity of ShareGNNs is lower-bounded by the discriminative power of the chosen invariants, giving a principled way to control model complexity and demonstrate that ShareGNN achieves competitive results on graph-level classification and regression tasks with only one message-passing layer.

## 2 MAIN APPROACH

At the heart of our approach lies a new paradigm for learning weights in neural networks. Instead of relying on fixed-size weight matrices, we treat all learnable parameters as elements of an unordered collection. For a given input graph, the relevant weights are dynamically selected and assembled into matrices or bias terms according to the chosen graph invariants. This on-the-fly construction of weight matrices allows the model to adaptively reuse parameters across structurally equivalent

parts of a graph and even across graphs, while maintaining full flexibility. Building on this idea, we instantiate two types of invariant-based weight sharing:

(a) *Encoder message passing*: weights are indexed by invariant signatures of node pairs (e.g., labels and distance), enabling information flow over arbitrary pairs.

(b) *Decoder pooling and bias*: weights depend on invariants of single nodes, allowing structure-aware pooling and bias terms.

In both cases, the model constructs the relevant weight structures dynamically from a shared parameter pool, guaranteeing permutation equivariance in the encoder and permutation invariance in the decoder.

## 2.1 PRELIMINARIES

We denote by $\mathbb{N}_0$ the set of natural numbers including zero. A *graph* is a pair $G = (V, E)$ where $V$ is the set of nodes and $E \subseteq \{\{i, j\} \mid i, j \in V\}$ is the set of edges. Each graph $G = (V, E)$ is associated with an initial node feature matrix $X^{(0)} \in \mathbb{R}^{|V| \times k}$, where $k \in \mathbb{N}$ is the node feature dimension. This matrix also fixes an ordering of nodes in $V$. For a node $i \in V$, its *neighborhood* is denoted by $\mathcal{N}_i := \{j \in V : \{i, j\} \in E\}$, and its *degree* is $\deg(i) := |\mathcal{N}_i|$. The *distance* between two nodes $i, j \in V$, i.e., the length of a shortest path between them, is written as $d(i, j)$. A *labeling function* $l : V \to \mathbb{N}_0$ assigns an integer label to each node.

## 2.2 INVARIANT-BASED WEIGHT SHARING

We now formalize the two types (a) and (b) of invariant-based weight sharing from above. In both cases, graph invariants provide indices into a shared parameter pool from which the corresponding weights are constructed.

**Weight Sharing for Message Passing** For type (a), we associate a learnable weight with each valid invariant triple that characterizes a pair of nodes. A triple consists of: (i) the label of the first node, (ii) the label of the second node, and (iii) the shortest-path distance between them. Thus, given a labeling function $l : V \to \mathbb{N}_0$, the triple for a node pair $(v, w) \in V \times V$ is defined as $\tau_{vw} := (l(v), l(w), d(v, w)) \in \mathbb{N}_0^3$. For molecular graphs, $l$ can assign to each node its atomic number, while for general graphs $l$ may, for example, assign the node's degree or any other structural identifier. Moreover, the labeling function for the source and target nodes could in principle be chosen differently, but we omit this distinction in our experiments for simplicity. To control the parameter budget, we restrict ourselves to a finite set of valid triples $\mathcal{T} \subset \mathbb{N}_0^3$. For each valid triple $\tau \in \mathcal{T}$, we assign a learnable scalar weight $\omega_\tau \in \mathbb{R}$. Node pairs with triples not in $\mathcal{T}$ get weight zero, ensuring sparsity and scalability. For example, if $\mathcal{T}$ includes triples with distances up to $D$, then any pair of nodes whose shortest-path distance exceeds $D$ will be assigned zero weight and will not exchange messages. Let $\Omega_{\mathcal{T}}$ be the set of weights associated with all valid triples. This defines the invariant mapping

$$\mathrm{m}_{\mathcal{T}}^l : V \times V \to \mathcal{T} \cup \{0\} \to \Omega_{\mathcal{T}} \cup \{0\} \tag{1}$$

that assigns to each pair of nodes $(v, w)$ a weight $\omega_{\tau_{vw}}$ if $\tau_{vw} \in \mathcal{T}$ and zero otherwise.

While we focus on label-label-distance triples for simplicity, this association is only one example: other invariants (such as edge labels, motif counts, or higher-order structural descriptors) can be incorporated in exactly the same way, see Appendix E for an example with edge labels.

**Weight Sharing for Bias and Pooling** For type (b), we assign shared weights to individual nodes rather than node pairs. Given a labeling function $l : V \to \mathbb{N}_0$ (not necessarily the same as before) and a set of valid labels $\mathcal{L} \subset \mathbb{N}_0$, each label $\lambda \in \mathcal{L}$ is associated with a learnable weight vector $\omega_\lambda \in \mathbb{R}^k$, where $k$ is the node feature dimension. The set of all such vectors is denoted by $\Omega_{\mathcal{L}, k}$. This yields

$$\mathrm{b}_{\mathcal{L}, k}^l : V \to \mathcal{L} \cup \{0\} \to \Omega_{\mathcal{L}, k} \cup \{0\} \tag{2}$$

which assigns each node $v \in V$ a learnable weight vector $\omega_{l(v)} \in \mathbb{R}^k$ if $l(v) \in \mathcal{L}$ and zero otherwise. During encoding, these vectors act as bias terms, while during decoding they are used as pooling weights to aggregate node embeddings. As with type (a), the choice of labeling function is flexible: for molecules, $l$ may correspond to atomic numbers, while for arbitrary graphs it may use node degree, Weisfeiler–Leman labels, or other structural identifiers. In the following, we describe several labeling functions that instantiate these ideas.

## 2.3 LABELING FUNCTIONS

The choice of labeling function $l$ determines how nodes (or node pairs) are grouped for parameter sharing. Depending on the application, these labels encode properties, such as atomic numbers in molecular graphs, or richer structural information, such as degrees, Weisfeiler–Leman (WL) labels, or existance of small patterns. In our experiments, we consider the following labeling strategies:

*Molecular Labeling.* For molecular graphs, $l$ assigns to each node its atomic number.

*Weisfeiler-Leman (WL) Labeling.* The WL algorithm iteratively refines node labels by hashing together a node's current label with the multiset of labels of its neighbors. After a fixed number of iterations, this procedure yields labels that capture information about the $i$-hop neighborhood around each node (Shervashidze et al., 2011). In our experiments, we use 1-WL labels with varying depths.

*Pattern Labeling.* In addition to WL labels, we consider labeling nodes by their participation in small subgraph patterns. For a given set of patterns (e.g., triangles, cliques, cycles), we count for each node how many embeddings of each pattern include that node. This count vector is then hashed into a unique integer label, providing additional structural distinctions beyond WL.

These strategies can also be combined, e.g., by concatenating the atomic number, WL label, and pattern labels into a single composite label. This combined labeling allows the model to capture multiple levels of structural information simultaneously.

## 2.4 INVARIANT-BASED LAYERS

We now describe how invariant-based weight sharing is integrated into the architecture of ShareGNNs. Each encoder and decoder layer dynamically assembles its weight matrices from a shared parameter pool according to the chosen invariants. Given an input graph, the corresponding invariant mappings are applied to its nodes (or node pairs) to get the appropriate weights.

**Invariant-Based Encoder** An invariant-based encoder layer consists of two components: (1) a message-passing matrix whose entries are determined by the type (a) invariant mapping $\mathrm{m}_{\mathcal{T}}^{l_1}(v, w)$, and (2) a bias matrix whose rows are determined by the type (b) mapping $\mathrm{b}_{\mathcal{L},k}^{l_2}(v)$. Unlike fixed weight matrices, these matrices are constructed dynamically for each input graph from the layer's parameter pool. Concretely, let $G = (V, E)$, with $|V| = n$ be the input graph and $X^{(h)} \in \mathbb{R}^{n \times k}$ the node embeddings after $h$ layers. Assume that the nodes $v, w \in V$ correspond to the positions $i, j$ in $X^{(h)}$, i.e., they are nodes $i, j$ of the fixed node ordering of $G$. Then, for each node pair $(v, w)$, the entry $W_{ij}$ of the message-passing matrix corresponds to the weight $\mathrm{m}_{\mathcal{T}}^{l_1}(v, w)$, where $l_1$ is the labeling function and $\mathcal{T}$ the valid set of triples. Similarly, the $i$-th row of the bias matrix is given by $\mathrm{b}_{\mathcal{L},k}^{l_2}(v)$. The encoder updates node embeddings as

$$\underbrace{X^{(h+1)}}_{n \times k} = \sigma \left( \underbrace{\mathbf{W} \left[ G, \mathrm{m}_{\mathcal{T}}^{l_1}, \Omega_{\mathcal{T}} \right]}_{n \times n} \cdot \underbrace{X^{(h)}}_{n \times k} + \underbrace{\mathbf{b} \left[ G, \mathrm{b}_{\mathcal{L},k}^{l_2}, \Omega_{\mathcal{L},k} \right]}_{n \times k} \right) \tag{3}$$

where $\sigma$ is a non-linear activation function. This formulation defines a form of message passing that: is not limited to adjacent nodes, since any valid invariant triple can link two nodes, allows directional effects, since $\mathrm{m}_{\mathcal{T}}^{l_1}(v, w)$ can differ from $\mathrm{m}_{\mathcal{T}}^{l_1}(w, v)$, and reuses the same weights for all pairs that share the same triple. As a result, a single encoder layer can propagate information over arbitrary distances while preserving permutation equivariance. Figure 4 (left) illustrates this mechanism for node $v_i$.

**Invariant-Based Decoder** The decoder aggregates the final node embeddings into a fixed-size graph representation using the type (b) mapping. Instead of mean or max pooling, it performs a weighted pooling where the contribution of each node is determined by its invariant label. Given the representation $X^{(h)} \in \mathbb{R}^{n \times k}$, we construct a pooling matrix $\mathbf{W}$ whose $i$-th column corresponds to the weight vector $\mathrm{b}_{\mathcal{L},m}^{l}(v_i) \in \mathbb{R}^m$ retrieved from the decoder's parameter pool $\Omega_{\mathcal{L},m}$. The aggregated representation is then computed as

$$\underbrace{\mathbf{X}^{(h+1)}}_{m \times k} = \sigma \left( \frac{1}{n} \underbrace{\mathbf{W} \left[ G, \mathrm{b}_{\mathcal{L},m}^{l}, \Omega_{\mathcal{L}} \right]}_{m \times n} \cdot \underbrace{\mathbf{X}^{(h)}}_{n \times k} + \underbrace{\mathbf{b}}_{m \times k} \right) \tag{4}$$

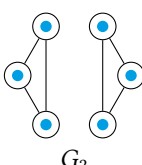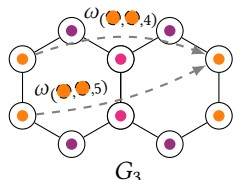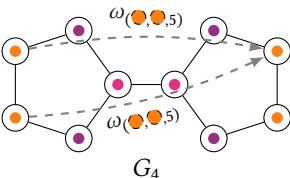

$G_1$          $G_2$          $G_3$          $G_4$

Figure 2: Non-isomorphic graph pairs $G_1, G_2$ and $G_3, G_4$ that are indistinguishable by the 1-WL test, and therefore by standard message-passing GNNs, but distinguishable by ShareGNN. Node colors indicate 1-WL labels. Arrows mark messages in ShareGNN that enable distinguishing the graphs.

where $m$ is the desired output dimension and $\mathbf{b}$ is a standard bias term of fixed size $m \times k$. The $1/n$ normalization mitigates the effect of graph size. This pooling strategy preserves permutation invariance and allows the decoder to emphasize structurally important nodes, as determined by the chosen invariant labels. Figure 4 (right) visualizes this pooling step.

**Multi-Heads** While encoder and decoder operate with a single set of invariants, ShareGNNs naturally extend to a multi-head architecture. Each encoder or decoder head is instantiated in parallel, with its own parameter pool and potentially different labeling functions. The heads are concatenated and combined by a feed-forward layer before proceeding to the next stage. This multi-head design allows the model to capture complementary structural patterns within the same layer, see Figure 11.

**Backpropagation** ShareGNNs use standard backpropagation. Once the dynamic weight matrices are constructed, gradient computation proceeds automatically through the computational graph.

## 2.5 SHAREGNNS

A ShareGNN is built by stacking one or more invariant-based encoder layers followed by a decoder that aggregates the final node embeddings into a graph-level representation. This design offers several advantages: *(i) Adaptivity:* The model is not constrained to fixed-size weight matrices; it dynamically adapts to graphs of varying sizes. *(ii) Expressiveness:* Its ability to capture structural patterns is determined by the choice of invariants and is not limited by standard neighborhood-based schemes. *(iii) Long-range interactions:* It propagates information across arbitrary node pairs within a single layer, avoiding over-smoothing and over-squashing. *(iv) Permutation awareness:* The encoder is permutation equivariant, and the decoder is permutation invariant by construction. *(v) Transferability and interpretability:* Since parameters are indexed by structural invariants, the learned weights can be transferred across different datasets, and each parameter has a clear structural meaning. Some current limitations of our approach is that the choice of graph invariants relies on domain knowledge or grid search which is standard practice for information encoding in GNNs. Learning invariants or the architecture directly from data is an exciting direction for future work. Moreover, our approach has so far been explored primarily for graph-level tasks; extending it to node classification, link prediction, and other domains (e.g., text or images) is an important next step. We now turn to a formal analysis of the properties of ShareGNNs (proofs in Appendix A).

**Proposition 1.** *The encoder layer as defined in Equation* (3) *is permutation equivariant, i.e., for each permutation $\pi$ of the node order of the input graph it holds $\pi(\mathbf{X}^{(h+1)}) = \sigma(\mathbf{W} \cdot \pi(\mathbf{X}^{(h)}) + \mathbf{b})$.*

**Proposition 2.** *The decoder as defined in Equation* (4) *is permutation invariant, i.e., for each permutation $\pi$ of the node order of the input graph it holds $\mathbf{X}^{(h+1)} = \sigma(\frac{1}{n}\mathbf{W} \cdot \pi(\mathbf{X}^{(h)}) + \mathbf{b})$.*

**Corollary 3.** *ShareGNNs are permutation invariant.*

**Proposition 4.** *Each ShareGNN is at least as expressive as the labeling function l of the decoder, i.e., if two graphs $G$ and $G'$ are distinguishable by l then they are also distinguishable by the ShareGNN.*

**Corollary 5.** *For non-isomorphic graphs $G$ and $G'$ it exists a ShareGNN distinguishing $G$ and $G'$.*

The results show that ShareGNNs are permutation invariant by construction and can be made arbitrarily expressive by choosing suitable invariants. Beyond labels, distance-based message passing further strengthens expressivity. For example, Figure 2 shows graph pairs that are indistinguishable

by the 1-WL test, and therefore by standard message-passing GNNs, but can be distinguished by ShareGNN using 1-WL node labels.

**Time and Space Complexity**    The runtime of ShareGNNs consists of two parts: (i) a preprocessing stage, where node labels and pairwise distances are computed, and (ii) the actual training and inference. The cost of computing node labels depends on the chosen invariants but is in our case negligible in practice compared to model training (Table 7). Pairwise distance computations require $O(n^2)$ time per graph and are computed only once for the entire dataset. For message passing, let $D$ denote the maximum propagation distance and $N$ the number of distinct node labels in the dataset. In the worst case, this yields at most $N^2 \cdot D$ parameters. In practice (cf. Table 8) the number is significantly smaller: not all label pairs occur at every distance, and weights unused in the dataset are discarded. If the label space is very large, $N$ can also be restricted explicitly. Assigning weights to node pairs requires $O(n^2)$ time and $O(n^2)$ space to store the mapping from node pairs to their corresponding weights. Overall, the asymptotic complexity is comparable to the attention mechanism in graph transformers, but with the added benefit of flexibility in reducing the number of parameters through the choice of invariants.

## 3    RELATED WORK

We position our approach in the context of prior work on weight sharing and graph neural networks.

**Weight sharing**    Weight sharing was first introduced in neural networks by Rumelhart et al. (1986) and became a cornerstone of convolutional networks (LeCun et al., 1989), where shared kernels provide translation invariance and parameter efficiency. Extending these ideas to irregular structures such as graphs has been challenging because graphs lack a regular grid structure.

**Message passing**    In message passing networks (Kipf & Welling, 2017; Gilmer et al., 2017; Hamilton et al., 2017) node embeddings are updated by aggregating information from neighbors. Variants such as GAT (Velickovic et al., 2017) and GATv2 (Brody et al., 2022) implement attention mechanisms to learn the importance of neighboring nodes. These models typically learn weights and importance between neighboring nodes based on node features.

**Beyond message passing**    Several approaches overcome the locality and limitations of standard message passing by introducing global interactions and richer structural information. These include graph transformers (Yun et al., 2019; Shi et al., 2021; Brody et al., 2022; Zhu et al., 2023; Buterez et al., 2024), which enable all node pairs to interact in a single layer through learned attention, as well as approaches based on subgraph-level reasoning (Zhang et al., 2023; Puny et al., 2023; Paolino et al., 2024; Yan et al., 2024), higher-order representations using simplicial and cellular complexes (Bodnar et al., 2021a;b), provably powerful architectures (Maron et al., 2019), or positional/distance encodings (Li et al., 2020; Kreuzer et al., 2021; Rampásek et al., 2022; Franks et al., 2025). However, the works still parameterize their operations in terms of edges, attention scores, or node features, without a principled mechanism for sharing parameters across structurally equivalent substructures.

In contrast, ShareGNN introduces a structurally driven weight-sharing mechanism. Rather than learning node-feature-based relations, the weights are indexed by invariants and can be reused within and across graphs as there is no dependence on some fixed (positional) encoding of nodes. We combine the connectivity of transformers with the message passing paradigm and weight sharing, resulting in a flexible architecture that is inherently interpretable with transferable weights.

## 4    EXPERIMENTS

We evaluate ShareGNN on benchmarks designed to test both expressiveness and predictive performance. Our experiments address three main questions: (1) Can invariant-based weight sharing recover theoretical properties such as substructure counting? (2) Does it achieve competitive performance on standard graph classification and regression benchmarks? (3) Can it do so with shallow architectures, highlighting the benefits of long-range message passing in a single layer? To answer these questions, we perform experiments on real-world molecular and social network datasets and synthetic benchmarks designed to evaluate generalization beyond local neighborhoods. We provide dataset details

|  | triangle | tailed triangle | star | 4-cycles | 5-cycles | 6-cycles |
|---|---|---|---|---|---|---|
| PPGN | 8.9 | 9.6 | 14.8 | 9.0 | 13.7 | 16.7 |
| GNN-AK+ | 12.3 | 11.2 | 15.0 | 12.6 | 26.8 | 58.4 |
| SUN (EGO+) | 7.9 | **8.0** | 6.4 | 10.5 | 17.0 | 55.0 |
| GNN-SSWL+ | **6.4** | 6.7 | 7.8 | **7.9** | **10.8** | **15.4** |
| **ours (one)** | 1.9 | 3.2 | 2.4 | 3.6 | 5.3 | 5.1 |
| **ours (all)** | 3.3 | 9.1 | 6.3 | 5.6 | 5.3 | 7.8 |

Table 1: Substructure Counting Benchmark MAE ($\downarrow$) in $10^{-3}$ on different tasks. The best results are highlighted by **First**, **Second** and **Third**.

(Appendix C.3), the full experimental setup including hyperparameter configurations (Appendix C), as well as extended results and ablation studies (Appendix D). Our code to reproduce the experiments is available at https://anonymous.4open.science/r/ShareGNN_ICLR_2026.

### 4.1 SUBSTRUCTURE COUNTING

The Substructure Counting Benchmark (cf. (Zhao et al., 2022; Frasca et al., 2022)) contains graphs labeled with counts of six motifs (triangles, tailed triangles, stars, and 4-6 cycles) and measures the ability to distinguish structural patterns. We train a ShareGNN with a single encoder and decoder layer, using five heads: four heads dedicated to specific cycle lengths and one capturing the global structure. We consider two settings: one separate model for each motif (one) and a single model jointly for all six motifs (all). Baselines include PPGN (Maron et al., 2019), GNN-AK+ (Zhao et al., 2022), SUN (EGO+) (Frasca et al., 2022), and GNN-SSWL+ (Zhang et al., 2023). Table 1 shows that ShareGNN achieves state-of-the-art results in the single-task setting and remains competitive in the multi-task setting, confirming the theoretical advantages of invariant-based weight sharing even with shallow architectures. Figure 11 visualizes the activations of the different heads, highlighting that ShareGNNs's predictions can be directly interpreted in terms of the structural motifs it uses.

### 4.2 CLASSIFICATION

We benchmark ShareGNN on diverse graph classification tasks, including molecular graphs and social graphs, along with synthetic datasets. Since standardized evaluation protocols are often missing in graph learning, direct comparison across works is difficult (Errica et al., 2020). To ensure clarity and reproducibility, we follow *two* widely used setups: *fair* (Errica et al., 2020) and *standard* (Xu et al., 2019). Full details of the results of the second setup are provided in Appendix C.5.

**Dataset Choice** As noted by, e.g., Schulz & Welke (2019); Errica et al. (2020); Bechler-Speicher et al. (2024), it difficult to assess the true benefits of advanced models, as even simple baselines often perform competitively. For very small datasets (MUTAG, PTC), we observed that results vary dramatically with the data split, making meaningful comparisons unreliable. To ensure a fair and informative evaluation, we therefore focus on datasets where structural information is essential by also including challenging synthetic benchmarks RingTransfer (RT1-RT3) and Snowflakes (SF). These are designed to test long-range dependencies and structural reasoning. Moreover, we consider molecular datasets (DHFR, Mutagenicity (Muta), NCI1, NCI109), social network datasets (IMDB-BINARY (IM-B), IMDB-MULTI (IM-M)) collected by Morris et al. (2020).

***Fair* Evaluation (Errica et al., 2020)** Each dataset is split into 10 predefined folds, with separate training, validation, and test sets. Models are trained on the training folds, and the hyperparameter configuration that performs best on the validation sets is used to evaluate the corresponding test sets. We report the average test accuracy over 10 folds, selecting the epoch with the best validation performance for each fold and repeating the procedure with three random seeds (Table 2).

**Competitor Choice** We consider several baselines, including, non-neural methods such as simple label histograms (NoG) (Schulz & Welke, 2019) and Weisfeiler–Leman subtree kernel (WL) (Shervashidze et al., 2011), classical GNNs, GraphSAGE (Hamilton et al., 2017), GIN (Xu et al., 2019),

| Method | NCI1 | NCI109 | Muta. | DHFR | IM−B | IM−M | RT1 | RT2 | RT3 | CSL | SF |
|---|---|---|---|---|---|---|---|---|---|---|---|
| NoG | 63.3° | 61.4° | 72.5° | 61.0★ | 70.7★ | 47.1• | 30.4• | 46.3• | 22.1• | 10.0° | 25.2• |
| WL | **85.2**• | 85.5° | 83.6° | 84.3° | 71.8★ | 51.9★ | 100.0° | 49.6★ | 24.7• | 10.0° | 25.7★ |
| GraphSAGE | 79.0• | 78.5° | 80.0° | 79.8• | 70.2• | 47.8• | 31.5★ | 50.1• | 26.7• | 10.0° | 23.3• |
| GIN | 80.3• | 79.1° | 81.7° | 80.0• | 67.3• | 44.9• | 31.7• | 51.2• | 26.2• | 8.7• | 24.2• |
| GAT | 76.1• | 75.4° | 78.7° | 78.4• | 68.0• | 47.8• | 33.0• | 59.4★ | 36.9★ | 10.0° | 25.4• |
| GATv2 | 80.4• | 78.6° | 78.8° | 78.3• | 69.8• | 48.0• | 33.4• | 63.9• | 35.4★ | 10.0° | 24.0★ |
| GT | 80.5° | – | – | – | 73.1° | 49.0∘ | – | – | – | – | – |
| GT + R−Cov | 83.1° | – | – | – | 76.1° | 51.1• | – | – | – | – | – |
| **ours** | 85.4° | 85.3° | 83.2° | 80.9★ | 75.9• | 51.1• | 100.0° | 100.0° | 99.9° | 100.0° | 98.0° |
| **ours (Random)** | 85.3• | 85.4° | 83.3° | 81.5• | 75.6• | 51.6★ | 100.0° | 100.0° | 96.9° | 100.0° | 98.0• |

Table 2: *Fair* evaluation (Accuracy in %). The best results are highlighted by **First**, **Second** and **Third**. The standard deviation is denoted by ° for small deviation ($0.0 - 2.0$), • for medium deviation ($2.1 - 4.0$) and ★ for large deviation ($> 4.0$).

| ZINC | GIN (Xu et al., 2019) | GSN (Bouritsas et al., 2023) | GNN-SSWL+ (Puny et al., 2023) | PPGN++ (6) (Zhang et al., 2023) | ESC-GNN (Yan et al., 2024) | **ours** |
|---|---|---|---|---|---|---|
| (12k) | 0.163 ± 0.004 | 0.101 ± 0.010 | 0.070 ± 0.005 | 0.071 ± 0.001 | 0.075 ± 0.002 | 0.107 ± 0.003 |
| (250k) | 0.088 ± 0.002 | – | 0.022 ± 0.002 | 0.020 ± 0.001 | 0.021 ± 0.003 | 0.024 ± 0.001 |

Table 3: ZINC Benchmark (MAE ↓). The best two results are highlighted by **First** and **Second**.

GAT (Velickovic et al., 2017) and GATv2 (Brody et al., 2022), and modern transformer-based models, GraphTransformer (GT) (Shi et al., 2021) and GT+R-Cov (Bechler-Speicher et al., 2024) that are evaluated in the *fair* setup. All baselines use the same train/validation/test splits, if available by Errica et al. (2020) and Dwivedi et al. (2023) (CSL).

**ShareGNN Configuration** Invariant-based weight sharing makes it unnecessary to encode node attributes explicitly. In these experiments, the initial node features are set to a constant scalar (1), and all structural information enters the model solely through the labeling functions. For robustness, we additionally consider a variant where Gaussian noise with standard deviation 0.5 is added to the constant inputs. The decoder output dimension is set to the number of classes, producing a graph-level representation of the correct size without any additional projection. Here, we use a single encoder layer with one head. The only exception is RT3 with two encoder layers. Our hyperparameters are the labeling functions $l_e$ and $l_d$ for the encoder and decoder and the set of valid triples $\mathcal{T}$, controlling the range of message passing. For $l_e$ and $l_d$, we perform a grid search over: atomic numbers, 1-WL labels with depths from 0 to 3, and pattern labels (e.g., cycles and cliques) which is standard practice for these datasets (cf. (Bouritsas et al., 2023)). The valid triples $\mathcal{T}$ are restricted by the maximum hop distance $D$, which we set to 6 for molecular datasets and to the maximum graph diameter for social datasets. Additional experiments (Figure 13) confirm that the number of weights can be reduced without loss of accuracy by pruning $\mathcal{T}$, e.g., by choosing only triples that occur at least $N$ times in the dataset. We use *tanh* for activation in all layers, cross-entropy loss and the Adam optimizer (Kingma & Ba, 2015) with a learning rate of 0.01 (real-world) and 0.1 (synthetic). All biases are initialized with 0 and all other weights with a constant value of 0.001 (real-world) or a uniform distribution $[-0.1, 0.1]$ (synthetic). Interestingly, therefore, the only random factor in the real-world experiments is the order of the input samples. Models are trained for 200 epochs with a batch size of 64 and early stopping if the validation accuracy does not improve for 25 epochs in the *fair* evaluation.

**Results** The experimental results demonstrate that ShareGNNs achieve competitive performance (first or second rank Tables 2 and 11) across both evaluation setups while relying on shallow architectures. The performance is superior to ordinary GNNs and competitive to WL and Transformer-based models, confirming that invariant-based weight sharing enables a single-layer model to capture complex structural dependencies that typically require deep architectures. Notably, on synthetic benchmarks such as RT and SF, we achieve accuracies exceeding 98%, even though these datasets are designed to challenge standard message-passing methods. On datasets not solvable by 1-WL (e.g., CSL and Snowflakes), ShareGNN consistently outperforms message-passing baselines. This demonstrates that weight-sharing using graph invariants improves ShareGNNs expressivity compared

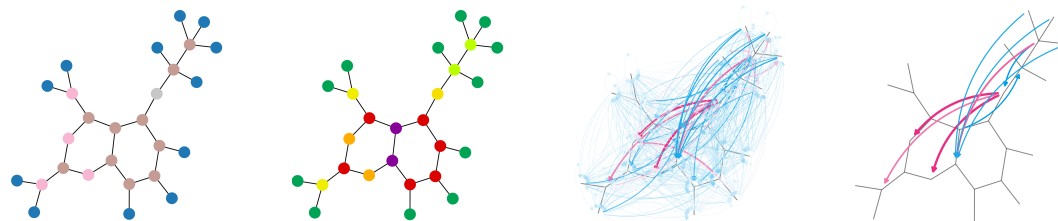

Figure 3: Invariant-based message passing (ShareGNN encoder layer) for a molecule from DHFR. Left to right: atomic numbers, graph invariant labels, learned weights, and the three largest positive (red), negative (blue) ones. The thickness and color of the edges scales with their absolute value.

to ordinary message passing. Additional experiments (Appendix D.2) in which the graph-invariant information from the best-performing ShareGNN configuration was provided as input features to competing models show that encoding information via weight sharing is more effective than encoding the same information as input features.

### 4.3 REGRESSION & SCALABILITY

To assess scalability, we evaluate ShareGNN on the ZINC benchmark dataset, considering both the $12k$ and $250k$ graph versions. For the experiments, we use a single-layer encoder and decoder with multiple heads. Each head focuses on a distinct type of structural invariant, see Appendix C.6. ShareGNNs attain competitive performance (MAE) on both datasets (Table 3) while maintaining a shallow architecture. Thus, our invariant-based design scales effectively to large datasets while retaining high accuracy, without requiring deep or complex networks. The remaining gap to the best-performing methods may be attributed to two factors: (i) the current encoding of edge labels, incorporated indirectly via WL labels, and (ii) several competing models leverage information beyond cycles, whereas ShareGNN focuses explicitly on information about cycles of lengths 6 to 10. Indeed, GSN (Bouritsas et al., 2023), also relying on cycle information, performs at a comparable level.

In summary, the experimental results demonstrate that ShareGNNs achieve competitive performance across diverse benchmarks. These findings confirm the effectiveness of invariant-based weight sharing as a principled mechanism for capturing structural information efficiently.

## 5 CONCLUDING REMARKS AND OUTLOOK

We introduced a new weight-sharing paradigm for graph structured data based on graph invariants, and demonstrated its effectiveness on graph learning through ShareGNN. The approach provides permutation-equivariant encoders and permutation-invariant decoders by construction, enabling long-range message passing in a single layer and avoiding the over-smoothing issues of deep architectures. Our analysis establishes a direct link between ShareGNN's expressivity and the discriminative power of the chosen invariants. Because invariants can be integrated without altering the architecture, Share-GNNs serve as a testbed for exploring new invariants and their impact on graph learning. Extensive experiments confirm that this paradigm achieves competitive or superior performance on substructure counting, synthetic, and real-world benchmarks, despite relying on remarkably shallow models.

A central advantage of invariant-based weight sharing is that weights are tied directly to the graph structure. Thus, the model offers an interpretation of which structures drive its predictions (cf. Figures 3, 11, 20 and 21). Moreover, weights are naturally transferable across datasets, e.g., weights learned on synthetic graphs might be reused for molecular graphs without modification. These properties make the invariant-based approach especially well suited for the emerging field of foundation models on graphs (Liu et al., 2023; Franks et al., 2025), as it naturally supports pretraining on large (synthetic) graph datasets followed by a transfer to new tasks. While our focus here was to establish the core formulation, analyze its theoretical properties, and evaluate its empirical performance, future work will scale this paradigm to larger and more diverse datasets, and investigate its role in transfer across domains and tasks.

## ETHICS STATEMENT

The authors declare that they have no potential conflicts of interest with respect to the research, authorship, and/or publication of this article. The work presented in this paper has not been published previously and is not under consideration for publication elsewhere.

## REPRODUCIBILITY STATEMENT

Our results can be reproduced using the code and instructions provided under https://anonymous.4open.science/r/ShareGNN_ICLR_2026. We provide the hardware specifications in Appendix C.2. All datasets used in our experiments are publicly available. We provide additional details in Tables 4 and 5 as well as illustrative examples of graphs from the synthetic datasets (Figures 5 to 9). Moreover, in the appendix we provide detailed information on the experimental setup, including hyperparameters (Tables 6, 8 and 10), preprocessing (Table 7) and architecture details (Table 9).

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

## THE USE OF LARGE LANGUAGE MODELS (LLMS)

We used large language models for minor editorial polishing (e.g., grammar, wording). All the ideas, analyses, and conclusions remain entirely our own.

## A  PROOFS

**Proposition 1** *The encoder layer is permutation equivariant, i.e., for each permutation $\pi$ of the node order of the input graph it holds $\pi(\mathbf{X}^{(h+1)}) = \sigma(\mathbf{W} \cdot \pi(\mathbf{X}^{(h)}) + \mathbf{b})$.*

*Proof.* We prove that the encoder layers of ShareGNNs are permutation equivariant, by showing that the following holds for all permutations $\pi$ of the nodes of the graph:

$$\pi\left(X^{(h+1)}\right) = \sigma\left(\mathbf{W} \cdot \pi\left(X^{(h)}\right) + \mathbf{b}\right)$$

Note that a permutation of the nodes of a graph corresponds to a permutation of the rows of its corresponding node representation $X^{(h+1)}$. The entries of the weight matrix and the bias term in our definition of the forward propagation depend on the fixed order of the nodes of the input graph. Thus, considering the permuted input $\pi(X^{(h)})$, the corresponding weight matrix and the bias term are by definition, permuted in the same way compared to the original input signal $X^{(h+1)}$. Hence, it follows that $\mathbf{W} \cdot \pi(X^{(h)}) = \pi(\mathbf{W} \cdot X^{(h)})$ and $\pi(\mathbf{W} \cdot X^{(h)}) + \mathbf{b} = \pi(\mathbf{W} \cdot X^{(h)} + \mathbf{b})$ which completes the proof.  $\square$

**Proposition 2** *The decoder layer is permutation invariant, i.e., for each permutation $\pi$ of the node order of the input graph it holds $\mathbf{X}^{(h+1)} = \sigma(\frac{1}{n}\mathbf{W} \cdot \pi(\mathbf{X}^{(h)}) + \mathbf{b})$.*

*Proof.* We prove that the decoder of ShareGNNs is permutation invariant, by showing that the following holds for all permutations $\pi$ of the nodes of the graph:

$$\mathbf{X}^{(h+1)} = \sigma\left(\frac{1}{n}\mathbf{W} \cdot \pi\left(\mathbf{X}^{(h)}\right) + \mathbf{b}\right)$$

Again, the columns of the weight matrix $\mathbf{W}$ depend on the fixed order of the nodes of the input graph. Thus, if the rows in $X^{(h)}$ are permuted, the corresponding columns of $\mathbf{W}$ are permuted in the same way. Hence, the result of the multiplication remains unchanged, i.e., we have that $\mathbf{W} \cdot \pi(\mathbf{X}^{(h)}) = \mathbf{W} \cdot \mathbf{X}^{(h)}$, which completes the proof.  $\square$

**Corollary 3** *ShareGNNs are permutation invariant.*

*Proof.* Permutation invariance of ShareGNNs follows directly from the permutation equivariance the encoder layers and the permutation invariance of the decoder.  $\square$

**Proposition 4** *Each ShareGNN is at least as expressive as the labeling function $l$ of the decoder, i.e., if two graphs $G$ and $G'$ are distinguishable by $l$ then they are also distinguishable by the ShareGNN.*

*Proof.* We show that the expressive power of the ShareGNNs is based on the expressive power of the underlying labeling functions. Indeed, we show that ShareGNNs are at least as powerful as the labeling function $l$ of the decoder. Assume, $l : V, V' \to \mathbb{N}_0$ is a labeling function that distinguishes the non-isomorphic graphs $G = (V, E)$ and $G' = (V', E')$ with $|V| = |V'| = n$ by comparing the histograms of the label counts. More precisely, if $G$ and $G'$ are non-isomorphic there exists some $a \in \mathbb{N}_0$ such that $A := \sum_{v \in V} l(v)=a \neq \sum_{v \in V'} l(v)=a =: A'$ with  being the indicator function. For example, we can define $l$ to be the $(k + 1)$-Weisfeiler-Leman labels with $k$ being the maximum of the treewidths of $G$ and $G'$ (Dvorák, 2010). Let ShareGNN consist of only a decoder layer based on the labeling function $l$ with a single output neuron, i.e., $m = 1$. The corresponding set of weights is denoted by $\Omega_{\mathcal{L}}$ where $|\mathcal{L}|$ is the number of different $(k + 1)$-Weisfeiler-Leman labels that occur for the nodes of $G$ and $G'$. Without loss of generality we assume that the graph representations of $G$ and $G'$ denoted by $X^G, X^{G'} \in \mathbb{R}^n$ are equal to vectors of ones. Let $\omega_a = 1$ and $\omega_b = 0$ for all $b \in \mathcal{L} \setminus \{a\}$. It follows that $\mathbf{W} \cdot X^G = A \neq A' = \mathbf{W} \cdot X^{G'}$ showing that the above defined ShareGNN is able to distinguish $G$ and $G'$. Moreover, we have shown that *every* ShareGNN is at least as powerful as the labeling function $l$ of the decoder.  $\square$

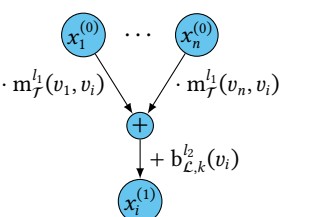 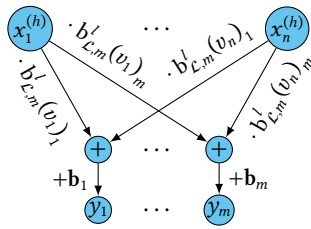

Figure 4: Update ($v_i$) in the encoder (left) and aggregation of the final node embeddings in the decoder (right)

**Corollary 5** *For each two non-isomorphic graphs $G$ and $G'$ there exists a ShareGNN that distinguishes $G$ and $G'$.*

*Proof.* The proof follows directly from Proposition 4 and the fact that there exists a labeling function $l$ that distinguishes $G$ and $G'$. For example, we can define $l$ to be the $(k + 1)$-Weisfeiler-Leman labels with $k$ being the maximum of the treewidths of $G$ and $G'$ (Dvorák, 2010). $\square$

## B   ARCHITECTURE VIZUALIZATIONS

Figure 4 visualizes the update of a single node $v_i$ in the encoder (left) and the aggregation of the final node embeddings in the decoder (right).

## C   EXPERIMENTAL DETAILS

We provide additional details about our experiments, including details on the benchmark datasets, the hyperparameters used to train the models, and the implementation of the ShareGNNs.

### C.1   IMPLEMENTATION

To reproduce the results we provide the implementation under the following link https://anonymous.4open.science/r/ShareGNN_ICLR_2026. Moreover, we give a detailed description on how to run the experiments, and how to add custom datasets, and custom labeling functions. Labeling functions and other hyperparameters are easily adaptable by changing our config files. In fact, all precomputation of already implemented labeling functions and pairwise distances between nodes is automated and also executed when custom datasets are added. Our implementation is fully compatible with the PyTorch Geometric library (Fey & Lenssen, 2019) and the PyTorch library (Paszke et al., 2019).

### C.2   RESOURCES

All experiments were conducted on a machine with an AMD Ryzen 9 7950X 16-core processor with 128 GB of RAM.

### C.3   DATASET DETAILS

In this section, we provide additional details about the datasets used in the experiments. First, we provide a detailed description of our new synthetic benchmark datasets.

***RingTransfer1***   The dataset consists of 1200 cycles of 100 nodes each, and is designed to test the ability to detect long-range dependencies. Four of the cycle nodes are labeled by $1, 2, 3, 4$, and all the others by $0$. The distance between each pair of the four nodes is exactly 25 or 50. The label of the graph is 0 if $d(1, 2) = 50$, 1 if $d(1, 3) = 50$, and 2 if $d(1, 4) = 50$. Figure 5 shows an example of the dataset. There are 400 graphs per class. The difficulty of the classification task lies in the fact that the information has to be propagated over a long distance. For ShareGNNs this is very easy because information can be propagated over arbitrary distances in a single encoder layer.

| Dataset | #Graphs | #Nodes | | | #Edges | | | Diameter | | | #Node Labels | #Classes |
|---|---|---|---|---|---|---|---|---|---|---|---|---|
| | | max | avg | min | max | avg | min | max | avg | min | | |
| NCI1 | 4 110 | 111 | 29.9 | 3 | 119 | 32.3 | 2 | 45 | 11.5 | 0 | 37 | 2 |
| NCI109 | 4 127 | 111 | 29.7 | 4 | 119 | 32.1 | 3 | 61 | 11.3 | 0 | 38 | 2 |
| Mutagenicity | 4 337 | 417 | 30.3 | 4 | 112 | 30.8 | 3 | 41 | 6.3 | 0 | 14 | 2 |
| DHFR | 756 | 71 | 42.4 | 20 | 73 | 44.5 | 21 | 22 | 14.6 | 8 | 9 | 2 |
| IMDB-BINARY | 1 000 | 136 | 19.8 | 12 | 1249 | 96.5 | 26 | 2 | 1.9 | 1 | 1 | 2 |
| IMDB-MULTI | 1 500 | 89 | 13.0 | 7 | 1467 | 65.9 | 12 | 2 | 1.5 | 1 | 1 | 3 |
| ZINC ($12k$) | 12 000 | 37 | 23.2 | 9 | 42 | 24.9 | 8 | 22 | 12.5 | 4 | 21 | - |
| ZINC ($250k$) | 249 456 | 38 | 23.2 | 6 | 45 | 24.9 | 5 | 23 | 12.5 | 3 | 28 | - |

Table 4: Details of the real-world datasets used in the experiments.

| Dataset | #Graphs | #Nodes | | | #Edges | | | Diameter | | | #Node Labels | #Classes |
|---|---|---|---|---|---|---|---|---|---|---|---|---|
| | | max | avg | min | max | avg | min | max | avg | min | | |
| RingTransfer1 | 1 200 | 100 | 100.0 | 100 | 100 | 100.0 | 100 | 50 | 50.0 | 50 | 5 | 3 |
| RingTransfer2 | 1 200 | 16 | 16.0 | 16 | 16 | 16.0 | 16 | 8 | 8.0 | 8 | 16 | 2 |
| RingTransfer3 | 1 200 | 16 | 16.0 | 16 | 16 | 16.0 | 16 | 8 | 8.0 | 8 | 16 | 4 |
| CSL | 150 | 41 | 41.0 | 41 | 82 | 82.0 | 82 | 10 | 6.0 | 4 | 1 | 10 |
| Snowflakes | 1 000 | 180 | 112.5 | 45 | 300 | 187.5 | 75 | 18 | 15.5 | 13 | 2 | 4 |
| Substructure Conting | 5 000 | 30 | 18.8 | 10 | 45 | 31.3 | 20 | 10 | 4.2 | 0 | 1 | - |

Table 5: Details of the synthetic datasets used in the experiments.

**RingTransfer2**   The dataset consists of 1200 cycles of 16 nodes each and is designed to test the ability to detect long-range dependencies as well as the ability to add node labels and detect even and odd numbers. The nodes in each graph are labeled from 0 to 15. For all nodes and their opposite node in the circle (distance 8) the sum of their labels is computed. If there are more even sums than odd sums, the graph is labeled 0, otherwise it is labeled 1. Figure 6 shows an example of the dataset. There are 600 graphs per class. We use the information that only distance 8 is relevant, by only assigning weights to node pairs with distance 8.

**RingTransfer3**   The dataset consists of the same graphs as *RingTransfer2*. However, the graphs are labeled differently. The graph label is determined by the labels of the nodes at distances 8 and 4 from the node with the label 0. We denote the node at distance 8 by $x$ and those at distance 4 by $y, z$. There are four cases: $x$ is even and $y + z$ is even, $x$ is even and $y + z$ is odd, $x$ is odd and $y + z$ is even, $x$ is odd and $y + z$ is odd. Each case corresponds to a class label from 0 to 3. Figure 7 shows an example of the dataset. We construct 300 graphs per class, that is, 100 graphs for each of the four cases. We use the information that information has to be collected from nodes at distances 8 and 4 only, by only assigning weights to node pairs with distances 4 and 8.

**Snowflakes**   The dataset consists of graphs proposed by Naik et al. (2024) that are indistinguishable by the 1-WL test, see Figure 9 for an example. The dataset consists of circles of length 3 to 12 and at each circle node a graph from $M_0, M_1, M_2$ or $M_3$ is attached, see Figure 10. $M_0, M_1, M_2$ and $M_3$ are non-isomorphic graphs that are not distinguishable by the 1-WL test. We refer to Naik et al. (2024) for the details. One node in the circle is labeled by 1 and all other graph nodes are labeled by 0. The label of the graph equals the index of the graph $M_0, M_1, M_2$ or $M_3$ that is attached to the circle node with label 1.

Tables 4 and 5 provide an overview of the real-world and synthetic datasets, including the number of graphs, the number of nodes, the number of edges, the diameter, the number of node labels and the number of classes.

## C.4   SUBSTRUCTURE COUNTING BENCHMARK

In this section, we provide additional details about the Substructure Counting Benchmark dataset and the configuration of the ShareGNNs. The Substructure Counting Benchmark dataset consists of 5 000 graphs (cf. Table 5) with predefined split 1 500/1 000/2 500 for training/validation/test. As for the other datasets the ShareGNN consists of one encoder layer and one decoder layer. Different from the classification benchmark configuration, we use multiple heads for both the encoder and decoder layers to capture different invariants in one layer. More precisely, for the encoder layer and

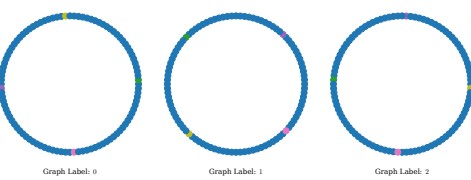

Figure 5: Example graphs taken from the *RingTransfer1* dataset.

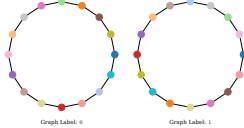

Figure 6: Example graphs taken from the *RingTransfer2* dataset.

decoder layer we use 5 heads each. The graph invariants respectively labeling functions used in the heads are simple cycles of length $3, 4, 5, 6$ and the degree of the nodes. In case of the encoder for the cycles we use only distance 0 information, corresponding to message-passing on self-connections and for the node degree we use all distances from 0 to 10, which is the maximum diameter of the graphs in the dataset. We concatenate the heads of the encoder using a feature output dimension of 10. The output dimension of the decoder is 10. We add a fully connected layer to match the required output dimension. All activations are LeakyReLU with the default slope of 0.01. The architecture definition can be found in the *network_substructure_counting.yml* configuration file. We use the Adam optimizer, MAE loss, batch size 64, constant learning rate 0.001 and 1000 epochs. Figure 11 shows the different activations per head. We see that for the cycle invariants only self-connection weights are set while using the degree as invariant we get weights between all pairs of nodes. Moreover, the visualization shows that for the different tasks (rows) the corresponding heads are activated while for the (all) model trained on all tasks simultaneously no specific head activation can be seen.

## C.5   CLASSIFICATION BENCHMARKS

In this section, we provide additional details about the preprocessing and training of the ShareGNNs for the graph classification benchmark datasets.

**Hyperparameter Search**   Each ShareGNN is defined by the labeling functions for the encoder layers and the decoder. As stated above we use no bias term for the encoder and only one encoder layer. Thus, in fact, our hyperparameters are one labeling function for the encoder (for message passing) and one for the decoder (for aggregation). Table 6 shows the hyperparameter grid used for our experiments on the real-world datasets, i.e., all labeling functions used for the encoder and the decoder. We differentiate between the original atomic numbers, node degrees, the Weisfeiler-Leman algorithm with different iteration depths, and subgraphs based on cycles and cliques of different lengths and sizes. We test 10 different options for the encoder and decoder layers for the molecules, and 22 and 18 options for the social datasets, respectively. Thus, in total we test 100 different hyperparameter configurations for the molecules and 396 for the social datasets. We did no hyperparameter search on the valid triples $\mathcal{T}$ as we used a fixed $D$ to determine the maximum distance between node pairs for which we compute shared weights. For the molecules we used $D = 6$, i.e., we compute shared message passing weights for all node pairs with distance at most 6 and for the social datasets we used $D = 2$. For the social datasets the maximum diameter is two, hence we define shared message passing weights for all node pairs in the graph. The best performing hyperparameter configuration for the *fair* evaluation setup is shown in Table 8. We note that our hyperparameter search is much faster than the one of the competitors in the *fair* evaluation setup using the implementation of Errica et al. (2020) and the same hardware, see also the note by Errica et al. (2020) regarding the runtime of their experiments.

**Preprocessing**   We need to precompute the node labels in advance for all labeling functions used in the hyperparameter search. Moreover, we need the pairwise distances between the nodes of the

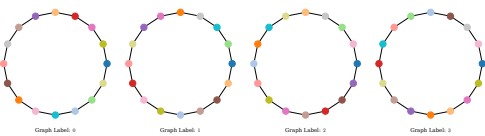

Figure 7: Example graphs taken from the *RingTransfer3* dataset.

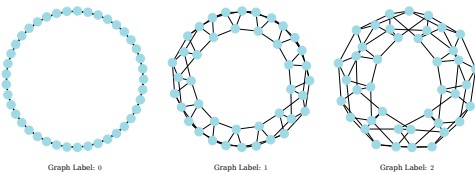

Figure 8: Example graphs taken from the *CLS* dataset.

graphs to compute all the valid triples. Table 7 shows the runtimes for the preprocessing of the datasets. All pairwise distances can be computed in less than 10 minutes, noting that we did not even pay attention to an efficient implementation (using networkx) as for these small datasets there was no need to do it. Except for the IMDB datasets the preprocessing time to compute the labels for all labeling functions for the hyperparameter search is negligible, again noting that we did not pay attention to an efficient implementation.

**The Best Hyperparameters**   Table 8 provides the details for the best hyperparameter configuration per dataset for the fair evaluation setup. The first column shows the average epoch in which the best validation accuracy was achieved, i.e., the epoch for which the test accuracy was calculated. It gives an indication of the model's convergence speed. In particular, for all real-world datasets the best validation accuracy was achieved on average after less than 50 epochs. This shows that our approach is very efficient and converges quickly. Recall, that the maximum number of epochs was set to 200. For NCI109, the best validation accuracy was already achieved on average after 3 epochs.

The second column shows the average time per epoch. The time per epoch depends on the number of parameters used in the model. Indeed, for NCI109 the best hyperparameter configuration of our model has approximately one million parameters and thus the time per epoch is higher than for the other datasets with about one minute per epoch. Particularly, except for the larger datasets NCI1, NCI109 and Mutagenicity the training time is less than 1 second per epoch. In general, runtime is not a problem for our approach as the preprocessing time is negligible and the training time is reasonable, compared to the competitors. Note that our computations run in parallel, i.e., we are able to run all the three runs and 10 folds in parallel on the same machine which produces some overhead but is more efficient than running the experiments sequentially. Indeed, implementing a dynamic approach in PyTorch is also a very challenging task, see also the comments in Han et al. (2022). They mention that there is a gap between the theoretical and practical runtime of dynamic neural networks because the implementation in PyTorch is not optimized for dynamic neural networks. This is also the reason why our approach runs approximately three times faster on the CPU than on the GPU.

Regarding the *Encoder Layers* in Table 8 we see which encoder labeling functions performed best for the different datasets. Notably, for the molecular datasets, Weisfeiler-Leman labeling with different iteration depths performed best. For the social datasets, the best results were achieved with labeling functions based using a combination of degree labeling and counting of patterns (triangles, squares).

For the *Decoder Layer* there is no uniform picture of which labeling functions performed best. The total number of parameters depends strongly on the chosen labeling functions. For the real-world datasets the number of parameters ranges from 10 183 in case of IMDB-BINARY to 925 956 in case of NCI109.

For the synthetic datasets we took only one hyperparameter configuration as we wanted to demonstrate that our approach is able to capture long-range dependencies and it is possible to restrict the message passing to a certain distance. For the RingTransfer1 dataset we use the original node labels and define shared weights only for self-connections and node pairs at distance 50. For RingTransfer2 we use the

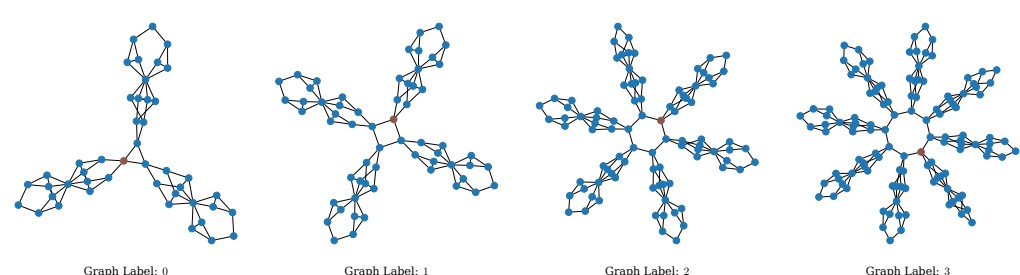

Graph Label: 0      Graph Label: 1      Graph Label: 2      Graph Label: 3

Figure 9: Example graphs from the *Snowflakes* dataset. The brown node in the circle is labeled by 1 and the other nodes by 0. The label of the graph is determined by the subgraph attached to the brown node.



Figure 10: Graphs $M_0, M_1, M_2$ and $M_3$ (Naik et al., 2024) that are not distinguishable by the 1-WL test.

original node labels and define shared weights only for node pairs with distance 8. For RingTransfer3 we use two encoder layers with the original node labels and define shared weights only for node pairs with distance 8 in the first layer and for node pairs with distance 4 in the second layer. For CSL we use node labels induced by patterns consisting of simple cycles up to length 10 and define shared weights only for self-connections and node pairs with distance 1. For the Snowflakes dataset we use combined node labels induced by patterns consisting of simple cycles of maximum length 10 and the original node labels. We define shared weights only for self-connections and node pairs with distance 3. In this way the ShareGNN is able to distinguish the graphs $M_0, M_1, M_2$ and $M_3$ that are not distinguishable by the 1-WL test. For the output layer we used the Weisfeiler-Leman with $i = 2$ iterations to collect the relevant information.

### C.6 REGRESSION BENCHMARK (ZINC)

For the ZINC benchmark, we conducted an extensive hyperparameter and architecture search over different invariant labeling functions, number of heads, maximum message-passing distances, and pooling strategies. The final configuration was selected based on the best validation performance. Table 9 summarizes the resulting architecture, and Table 10 lists the corresponding training hyperparameters.

The model consists of a single linear projection layer followed by an *invariant-based convolution layer* with multiple parallel heads. Each head corresponds to a specific structural invariant: induced/simple cycles (lengths 6–10), atomic numbers, and Weisfeiler–Leman labels with and without considering edge labels, all combined with distance features up to 23 hops (cycles and atomic numbers 23 hops and WL 2 hops) These heads enable parameter sharing across nodes and node pairs with the same invariant signature, allowing information to be propagated over the entire molecule in a single message-passing layer. The encoder heads are aggregated into a 100 dimensional representation for each node. A multi-head invariant pooling layer aggregates the resulting node embeddings into a graph representation, which is processed by a compact MLP (2 layer) readout. The invariant pooling layer uses the same invariants as the encoder layer but 10 heads per invariant instead of 2 heads (in the encoder).

| Molecules: Encoder/Decoder Labeling Function | Social: Encoder Labeling Function | Social: Decoder Labeling Function |
|---|---|---|
| Atomic Numbers
Node Degree | Node Degree | Node Degree |
| WL, Iterations 1
WL, Iterations 2
WL, Iterations 3 | Induced Cycles, Max. Length 10
Induced Cycles, Max. Length 20
Induced Cycles, Max. Length 4
Induced Cycles, Max. Length 5
Simple Cycles, Max. Length 3
Simple Cycles, Max. Length 4
Simple Cycles, Max. Length 5 | Induced Cycles, Max. Length 10
Induced Cycles, Max. Length 20
Induced Cycles, Max. Length 4
Induced Cycles, Max. Length 5
Simple Cycles, Max. Length 3
Simple Cycles, Max. Length 4
Simple Cycles, Max. Length 5 |
| Induced Cycles, Max. Length 10 + Atomic Numbers
Induced Cycles, Max. Length 20 + Atomic Numbers
Simple Cycles, Max. Length 10 + Atomic Numbers
Simple Cycles, Max. Length 20 + Atomic Numbers
Simple Cycles, Max. Length 6 + Atomic Numbers | Clique, Max. Size 3
Clique, Max. Size 4
Clique, Max. Size 10
Clique, Max. Size 20
Clique, Max. Size 6
Clique, Max. Size 50 | Clique, Max. Size 10
Clique, Max. Size 20
Clique, Max. Size 3
Clique, Max. Size 4
Clique, Max. Size 50
Clique, Max. Size 6 |
| | Clique Size 4
Squares + Node Degree
Triangles + Node Degree
Triangles, Squares + Node Degree | WL, Iterations 1
WL, Iterations 2
WL, Iterations 3
WL, Iterations 4 |
| | WL, Iterations 1
WL, Iterations 2
WL, Iterations 3
WL, Iterations 4 | |

Table 6: Labeling functions used for the encoder and decoder layers for the molecules (left) and social datasets (middle and right). For the molecules are 10 different options for the encoder and decoder layers, i.e., in total we tested 100 different hyperparameter configurations. For the social datasets there are 22 options for the encoder and 18 for the decoder layers, i.e., in total we tested 396 different hyperparameter configurations.

Table 7: Preprocessing times in seconds for the datasets used in the experiments. *Preprocessing Distances* shows the time needed to compute all the pairwise distances between the nodes of the graphs, *Preprocessing Labels* shows the time needed to compute the node labels for all node labeling functions used in the hyperparameter search.

| Dataset | Preprocessing Distances (s) | Preprocessing Labels (s) |
|---|---|---|
| NCI1 | 33.6 | 9.0 |
| NCI109 | 33.2 | 8.9 |
| Mutagenicity | 36.0 | 7.6 |
| DHFR | 11.4 | 1.8 |
| IMDB-BINARY | 4.9 | 1007.7 |
| IMDB-MULTI | 3.4 | 1618.9 |
| ZINC ($12k$) | 94.4 | 26.1 |
| ZINC ($250k$) | 1071.8 | 550.1 |
| RingTransfer1 | 94.9 | 0.0 |
| RingTransfer2 | 2.6 | 0.0 |
| RingTransfer3 | 2.7 | 0.0 |
| CSL | 2.3 | 5.0 |
| Snowflakes | 125.8 | 5.1 |
| Substructure Counting | 16.4 | 20.0 |

All experiments are trained with MAE loss and the Adam optimizer, using learning rate scheduling and weight initialization as specified in Table 10. The detailed layer and head definitions are provided in the configuration files `network_ZINC.yml` and `network_ZINC_full.yml`.

| Dataset | Best Epoch | Time per Epoch (s) | Encoder Layers | | | Decoder Layer | | # Total Weights |
|---|---|---|---|---|---|---|---|---|
| | | | Labeling Function | Distances | # Weights | Labeling Function | # Weights | |
| NCI1 | 6.4 ± 5.1 | 13.7 ± 9.1 | WL, Iterations 2 | 0, …, 6 | 381 145 | WL, Iterations 2 | 8 118 | 389 263 |
| NCI109 | 3.0 ± 5.0 | 67.2 ± 25.2 | WL, Iterations 3 | 0, …, 6 | 925 878 | Atomic Numbers | 78 | 925 956 |
| Mutagenicity | 14.7 ± 7.3 | 2.1 ± 0.5 | WL, Iterations 1 | 0, …, 6 | 28 638 | Atomic Numbers | 30 | 28 668 |
| DHFR | 41.3 ± 28.9 | 0.4 ± 0.1 | WL, Iterations 2 | 0, …, 6 | 53 715 | Simple Cycles, Max. Length 20 + Atomic Numbers | 292 | 54 007 |
| IMDB-BINARY | 21.9 ± 16.2 | 0.4 ± 0.1 | Triangles, Squares + Node Degree | 0, 1, 2 | 19 332 | Clique, Max. Size 4 | 64 | 19 396 |
| IMDB-MULTI | 15.6 ± 11.2 | 0.7 ± 0.2 | Triangles, Squares + Node Degree | 0, 1, 2 | 10 003 | Node Degree | 180 | 10 183 |
| RingTransfer1 | 200.0 ± 0.0 | 0.6 ± 0.1 | Node Labels | 0, 50 | 18 | Node Labels | 18 | 36 |
| RingTransfer2 | 200.0 ± 0.0 | 0.5 ± 0.1 | Node Labels | 8 | 240 | Node Labels | 34 | 274 |
| RingTransfer3 | 152.5 ± 56.8 | 0.7 ± 0.2 | *Layer 1:* Node Labels | 8 | 240 | Node Labels | 68 | 548 |
| | | | *Layer 2:* Node Labels | 4 | 240 | | | |
| CSL | 200.0 ± 0.0 | 0.0 ± 0.0 | Simple Cycles, Max. Length 10 | 0, 1 | 550 | Simple Cycles, Max. Length 10 | 960 | 1 510 |
| Snowflakes | 71.8 ± 44.7 | 0.5 ± 0.1 | Simple Cycles, Max. Length 10 + Node Labels | 0, 3 | 5 000 | WL, Iterations 2 | 36 | 5 036 |

Table 8: Details of the best performing hyperparameter configurations for ShareGNNs on the datasets used in the experiments. *Best Epoch* denotes the average epoch where the highest validation accuracy is reached, *Time per Epoch* is the average time per epoch in seconds, *Encoder Layers* shows the *Labeling Function* and *Distances* between node pairs used to define the shared weights. *#Weights* is the resulting number of weights. *Decoder Layer* shows the *Labeling Function* used to define the shared weights and *#Weights* is the resulting number of weights. *#Total Weights* is the total number of weights in the model.

| Layer | Configuration |
|---|---|
| Linear Layer | one-hot → 10 output features |
| Layer Norm | |
| Invariant-Based Message-Passing Layer | (each head twice, biases: *atomic numbers*) |
| | *induced cycles* (lengths: 6 to 10, $d = 1, …, 22$) |
| | *simple cycles* (lengths: 6 to 10, $d = 1, …, 22$) |
| | *atomic numbers* ($d = 1, …, 22$) |
| | *atomic numbers* ($d = 1$, edge labels) |
| | *degree + atomic numbers* ($d = 1, …, 4$) |
| | *degree + atomic numbers + edge labels* ($d = 1, …, 4$) |
| | *WL, depth: 1 + edge labels* ($d = 1, …, 2$) |
| Multi-Head Aggregation | 100 output features |
| Layer Norm | |
| Invariant-Based Pooling Layer | (same invariants but 10 heads per invariant) |
| Readout MLP | $100 \rightarrow 100 \rightarrow 1$ |

Table 9: Best ShareGNN architecture for the ZINC dataset. Invariants in italic. All layers except for the last one have LeakyRelu activations.

# D  ABLATION STUDY & ADDITIONAL RESULTS

We provide some additional results, including visualizations of the message-passing weights. In particular, we study the effect of restricting the number of weights as well as considering multiple encoder layers and different maximum distances for the node pairs.

## D.1  RESULTS FOR THE STANDARD EVALUATION

We already presented the results of our ShareGNNs regarding the fair evaluation protocol in Table 2. Here, we provide the results for the standard evaluation protocol in Table 11. We use the same splits as provided by Xu et al. (2019). In the *standard* setup, we compare to methods that explicitly enhance expressiveness through substructures or topological information, such as CIN (Bodnar et al., 2021a), SIN (Bodnar et al., 2021b), GSN (Bouritsas et al., 2023), PIN (Truong & Chin, 2024).

***Standard* Evaluation**  For comparison with Xu et al. (2019); Bodnar et al. (2021a); Bouritsas et al. (2023) we also adopt the widely used standard protocol. Each dataset is split into 10 predefined train/test folds. Models are trained on the training folds with a grid of hyperparameter settings, and the configuration with the best average test accuracy across folds is used for final reporting (Table 11).

| Hyperparameter | Value |
|---|---|
| Loss | Mean Absolute Error (MAE) |
| Optimizer | Adam + ReduceLROnPlateau (factor 0.5, patience 5) |
| Learning rate | 0.001 (min: 0.0001) |
| Batch size | 128 |
| Epochs | 150 |
| Weight & bias initialization | Uniform $[-0.001, 0.001]$ |
| Input features | One-hot node labels |
| Precision | double |

Table 10: Training hyperparameters for ZINC regression.

| | NCI1 | NCI109 | IMDB-B | IMDB-M |
|---|---|---|---|---|
| NoG | $63.4 \pm 2.5$ | $61.6 \pm 2.7$ | $71.2 \pm 5.6$ | $47.1 \pm 3.3$ |
| WL | $85.4 \pm 2.3$ | $85.5 \pm 1.6$ | $74.3 \pm 3.5$ | $51.5 \pm 3.9$ |
| CIN | $83.6 \pm 1.4$ | $84.0 \pm 1.6$ | $75.6 \pm 3.7$ | $52.7 \pm 3.1$ |
| SIN | $82.7 \pm 2.1$ | - | $75.6 \pm 3.2$ | $52.4 \pm 2.9$ |
| GSN | $83.5 \pm 2.3$ | - | $77.8 \pm 3.3$ | $54.3 \pm 3.3$ |
| PIN | $85.1 \pm 1.5$ | $84.0 \pm 1.5$ | $76.6 \pm 2.9$ | - |
| **ours** | $86.1 \pm 2.4$ | $86.8 \pm 1.6$ | $77.7 \pm 2.8$ | $53.1 \pm 4.0$ |

Table 11: *Standard* eval. (Acc. in %). The best results are highlighted by **First**, **Second** and **Third**.

**Results**  In the standard evaluation (Table 11), the performance matches or surpasses current state-of-the-art methods by noting that GSN, the best performing model on the social datasets, uses similar structural information. The ability to perform well in both protocols highlights the model's strengths and stability across different training setups. A variant using Gaussian noise added to the constant node features often shows slightly improved generalization, emphasizing that the model relies entirely on structure rather than node features.

### D.2 INFLUENCE OF INFORMATION ENCODING

Our approach encodes additional structural information via invariance-based weight sharing, i.e., all information is encoded in the message-passing weights of the encoder layer and the weights of the decoder layer. What should be analyzed is whether the improved performance of our ShareGNNs is solely due to the additional information used or whether the structural encoding of information is key to the performance. The following experiment shows that the additional information is not solely responsible for our improved performance but how the information is encoded matters. To do so, we collect the node labels of the best run of ShareGNNs and use them as additional input features for the competitors (Table 12).

Surprisingly, only GIN (on IMDB) and GraphSAGE (on DHFR) show a slight improvement compared to their original results. The overall performance is still worse than that of ShareGNN. We argue that the structural encoding of information is key to GNNs' performance and cannot be replaced by additional input features. Second, we vary the number of encoder layers and the maximum message-passing distance $D$ between two nodes. Figure 13a shows that with increasing distance $D$ the performance on NCI1 slightly improves from 82.9% ($D = 1$) to 85.6% ($D = 12$) for a single encoder layer. In fact, *a single* message-passing layer is sufficient, as it already yields the best performance. This shows that capturing long-range interactions in a single layer is better than capturing them in multiple layers. Third, we analyze the effect of valid triples $\mathcal{T}$ on the model performance. In particular, we reduce the number of shared message passing weights as follows: We consider only such shared weights for message passing that appear at least $x$-times in the graph dataset, i.e., we ignore rare relations. Figures 13b and 13c show the results for NCI1 and IMDB-B, respectively. Each $x$-axis value represents the model where all shared weights for message passing that occur at most $(x - 1)$-times in the respective graph dataset are removed. The performance on NCI1 is relatively

|  | NCI1 | DHFR | IMDB-B | IMDB-M |
|---|---|---|---|---|
| GraphSAGE | 0.4 | 2.3 | −1.1 | −1.9 |
| GIN | −2.8 | 0.4 | 3.4 | 3.0 |
| GAT | −1.1 | −0.4 | 0.9 | −0.7 |
| GATv2 | 0.2 | −0.9 | 0.0 | −0.5 |

Table 12: GNNs with additional node features (Mean change in accuracy in %), positive and negative changes. The GNNs use the same label information of the best run of ShareGNNs as additional input features.

stable up to an $x$-value of 8, i.e., the model considers only shared weights that occur at least 9 times in the NCI1 dataset. For IMDB-BINARY we see no significant drop in performance up to an $x$-value of 20. Thus, for both datasets, we can reduce the number of weights by more than 80% without any significant performance loss. We provide more details on the above ablation study for all the datasets in Appendix D.

### D.3 WEIGHT ANALYSIS OF THE ENCODER

Figure 14 shows the number of occurrences of the shared weights of the encoder layer summed over all graphs for the respective dataset. The underlying hyperparameter configuration, i.e., labeling function is the one with the best performance for the respective dataset, see Table 8. Each $x$-axis value corresponds to a weight of the encoder layer sorted by the number of occurrences. The $y$-axis shows the number of occurrences of the respective weight. The ten vertical dotted lines indicate the distribution of the number of occurrences. All weights in the plots that are right of the $n$-th vertical line (counting from the right) are present less than $(n + 1)$-times in the dataset, e.g., weights left of the leftmost vertical line appear more than 10 times in the dataset. The different colors denote the node pair distance the respective weight is associated with. We observe that for NCI and NCI109, more than half of the weights only occur once in the whole dataset. In fact, for all datasets only a small fraction of the weights occur more than 10 times. Figure 15 shows the final values of the trained encoder layer weights of the respective datasets. For all datasets we observe a gap in the distribution of the weights, which is best visible for the NCI109 dataset. We explain this behavior as follows: The message-passing weights are all initialized with the same constant value of 0.001. All weights are updated during training, except for the ones that are associated with node pairs that are only present in the test datasets. These weights are not updated and remain at their initial value. Weights that occur only a few times or even only once in the dataset are also less likely to be updated during training. Surprisingly, it seems that if the weights are updated, they are all updated above or below a certain threshold which results in the above-mentioned gap in the distribution of the weights. What is also surprising is that the weight values seem to be distributed symmetrically around zero, except of the IMDB-BINARY dataset.

### D.4 MULTI-LAYER SHAREGNNS

We study the effect of using multiple message-passing layers on the performance of the model as well as changing the maximum distance between two nodes for which message-passing weights are learned. Figure 17 shows the mean accuracy for ShareGNN for different numbers of encoder layers ($x$-axis) and different maximum distances between two nodes for which message-passing weights are learned ($y$-axis) for the NCI109 and DHFR datasets. Note that the models are initialized with the best hyperparameters for the respective dataset. We observe that including more distant node pairs for which message-passing weights are learned does not lead to a decrease in performance. Thus, long-range interactions in a single message-passing layer does not reduce performance. In contrast, if capturing long-range interactions with more and more encoder layers, the performance decreases. Indeed, the model is unable to fit the data with more than 4 encoder layers. This shows that capturing long-range interactions in a single layer surpasses the performance of capturing them in multiple layers.

### D.5 REDUCTION OF THE MESSAGE-PASSING WEIGHTS

The above observations motivate the following ablation study. The idea is to restrict the number of weights by considering only the most frequent weights that appear in the graph dataset Figure 18, and only the most infrequent weights that appear in the graph dataset Figure 19. Recall, that each shared weight in the encoder layer is associated with a node pair at a specific distance. Thus, we can pre-compute the number of occurrences of each weight and sort them accordingly for each dataset. Figure 18 shows the mean accuracies with standard deviation on the NCI109, DHFR and IMDB-MULTI datasets for models with different numbers of weights. Each $x$-axis value corresponds to a model that uses only such shared weights for message passing that occur at least $x$ times in the dataset. For DHFR and IMDB-MULTI, we observe that the performance only slightly decreases when deleting more and more of the less frequent weights. In this way we can reduce the number of weights by approximately 80% without a significant loss in performance. For the NCI109 dataset, the performance decreases more rapidly, likely because the number of parameters also drops quickly. Figure 19, where each $x$-axis value corresponds to a model that uses only such shared weights for message passing that occur at most $x$ times in the dataset, shows the opposite case. That is, we reduce the number of weights by removing the most frequent ones. We observe that the performance increases when adding more and more of the less frequent weights. However, the performance is still lower than the one of the models that use the most frequent weights despite the fact that the total number of weights is higher. We conclude that infrequent weights are less important for the graph classification task than the frequent ones.

Figure 16 visualizes the same as Figure 15 but this time for the models that are trained with message-passing weights that occur at least 10 times in the dataset. We see only one vertical dotted line in the plots. Weights to the right of the vertical line appear exactly 10 times; those to the left appear more than 10 times. Moreover, by comparing the $x$-axes of Figure 15 and Figure 16 we observe we train on a significantly smaller number of weights. Interestingly, the gap in the distribution of the weights values observed in Figure 15 has disappeared. This suggests that all considered weights are often enough updated during training. Another surprising observation is that in case of IMDB-BINARY, the weights seem to take certain levels. We can only speculate that these weights are associated with node pairs that are present in only one or a few graphs in the dataset, and thus are updated very infrequently during training. In fact, this would mean that they might not be that important for the graph classification task. An exact explanation for this behavior needs a deeper analysis of these weights and hence the corresponding node pairs. Despite the fact that we only consider weights that occur at least 10 times in the dataset, lots of weights in the IMDB-BINARY dataset seem to only occur once in the test dataset (bar around zero).

### D.6 VISUALIZATION OF THE MESSAGE-PASSING WEIGHTS

Our method allows for a very natural visualization of the learned message-passing weights and hence we expect our model to provide helpful insights regarding interpretability and explainability of graph classification tasks. Here are a few examples to illustrate this point. Figure 20 shows the learned weights of the encoder layers of our ShareGNN that achieves 100% accuracy on the RingTransfer2 and 99.4% accuracy on the RingTransfer3 dataset, respectively. Indeed, it is easy to observe that we restricted to such node pairs that are at distance 8 for RingTransfer2 (20a). For RingTransfer3 (20b), we see that the weights in the first layer are restricted to node pairs at distance 8 and in the second layer to node pairs at distance 4. Moreover, for RingTransfer3, we have rotated the graphs such that the node with the label 0 is at rightmost position for all three graphs. Recall, that the node with label 0 together with the labels of the nodes at distances 4 and 8 determine the label of the graph. The columns *Layer* 1 *Top* 3 and *Layer* 2 *Top* 3 in Figure 20b show the top three absolute weights of the respective encoder layers. In particular, we see how the model aggregates the necessary information from the nodes at distances 8 in the first layer, i.e., the largest weights are assigned to the node pairs containing the node with label 0. In the second layer, the model aggregates the information from the nodes at distances 4 and assigns the largest weights to the node pairs containing the node with label 0. Figure 21 shows the learned weights of the encoder layers of our ShareGNN for the CSL and Snowflakes datasets. In case of Snowflakes, comparing the columns *Node Labels* and *Top* 3 *Weights* in Figure 21b we observe that the model learns to aggregate the relevant information from the subgraphs $M_0, M_1, M_2$ and $M_3$ attached to the node with label 1. This validates that the model learns to look at the relevant subgraphs to classify the Snowflake graphs.

# E    EXTENDED EXAMPLE: EDGE LABELS

In this example we show that also edge labels can be included into the ShareGNN framework.

We consider the molecular graphs of ethylene and cyclopropenylidene, see Figure 22. The atoms of the molecules are denoted by $H$ for hydrogen and $C$ for carbon. The graph nodes correspond to the atoms and the graph edges to the atom bonds. Ethylene is represented by the initial vector $x \in \mathbb{R}^6$ and cyclopropenylidene by $y \in \mathbb{R}^5$. The bond types (*single* and *double*) can be seen as edge labels. Extending the considerations of our approach we show how to include edge labels into our model. Instead of considering for each pair of nodes a triple consisting of the atom labels and the distance between the nodes we consider triples consisting of the atom labels and the bond type between the atoms, i.e., each pair of nodes $(v, w)$ is associated with a triple $(l(v), l(w), \mathrm{bond}(v, w))$ where $l(v), l(w)$ is the atom label and $\mathrm{bond}(v, w)$ is either $\varnothing$ (no bond), $\odot$ (self-connection), $-$ (single bond) or $=$ (double bond). Triples $\mathcal{T}$ are valid if they do not contain $\varnothing$ in the third entry, i.e., it they are not of the form $(\cdot, \cdot, \varnothing)$. Moreover, if $\mathrm{bond}(v, w) = \odot$, i.e., the triple corresponds to a self-connection, then the the first two entries have to be equal. Finally, we assign each valid triple a learnable parameter from $\Omega_{\mathcal{T}}$. For our example, we need the following six learnable parameters:

$$\omega_1 := \omega_{(H,H,\odot)} \quad \omega_2 := \omega_{(C,C,\odot)} \quad \omega_3 := \omega_{(H,C,-)}$$
$$\omega_4 := \omega_{(C,H,-)} \quad \omega_5 := \omega_{(C,C,-)} \quad \omega_6 := \omega_{(C,C,=)}$$

Regarding the decoder, we use the atomic numbers for $l$ and fix the output size to $m = 2$. The set of valid labels $\mathcal{L}$ are all possible atomic numbers. Let $\Omega_{\mathcal{L}}$ be the set of learnable weight vectors for the decoder. For our example we need the following learnable parameters $\omega_H, \omega_C \in \mathbb{R}^2$.

The order of the graph nodes is fixed as shown in Figure 22. Recall, that we need the fixed order to construct the matrices for forward propagation matrices. Note that in this example we omit all bias terms for simplicity.

Using the learnable parameters defined above we get the following two matrices that define the forward propagation (message passing) of the encoder layer for the ethylene graph (left) and the cyclopropenylidene graph (right).

$$W_x^{\mathrm{enc}} = \begin{pmatrix} \omega_1 & 0 & 0 & 0 & \omega_3 & 0 \\ 0 & \omega_1 & 0 & 0 & \omega_3 & 0 \\ 0 & 0 & \omega_1 & 0 & 0 & \omega_3 \\ 0 & 0 & 0 & \omega_1 & 0 & \omega_3 \\ \omega_4 & \omega_4 & 0 & 0 & \omega_2 & \omega_5 \\ 0 & 0 & \omega_4 & \omega_4 & \omega_5 & \omega_2 \end{pmatrix} \quad W_y^{\mathrm{enc}} = \begin{pmatrix} \omega_1 & 0 & \omega_3 & 0 & 0 \\ 0 & \omega_1 & 0 & \omega_3 & 0 \\ \omega_4 & 0 & \omega_2 & \omega_6 & \omega_5 \\ 0 & \omega_3 & \omega_6 & \omega_2 & \omega_5 \\ 0 & 0 & \omega_5 & \omega_5 & \omega_2 \end{pmatrix}$$

We get the decoder weight matrices for the ethylene graph (left) and the cyclopropenylidene graph (right) as follows:

$$W_x^{\mathrm{dec}} = \begin{pmatrix} \omega_H & \omega_H & \omega_H & \omega_H & \omega_C & \omega_C \end{pmatrix} \quad W_y^{\mathrm{dec}} = \begin{pmatrix} \omega_H & \omega_C & \omega_C & \omega_H & \omega_H \end{pmatrix}$$

Combining the encoder and decoder layers we obtain

$$x' = \sigma^{\mathrm{dec}}(W_x^{\mathrm{dec}} \cdot \sigma^{\mathrm{enc}}(W_x^{\mathrm{enc}} \cdot x))$$

for the ethylene graph, and

$$y' = \sigma^{\mathrm{dec}}(W_y^{\mathrm{dec}} \cdot \sigma^{\mathrm{enc}}(W_y^{\mathrm{enc}} \cdot y))$$

for the cyclopropenylidene graph.

Note that the forward propagation of the encoder layer is essentially a multiplication with a weighted adjacency matrix of the graph, where the weights of the adjacency matrix are given by the learnable parameters (Figure 23). In contrast to adjacency matrices, the weight matrix is not necessary symmetric. The computation graph induced by the weight matrices exactly represent the graph structure while the edge weights are shared across the network (Figure 23).

This example uses edge labels only for message passing between adjacent nodes. However, edge label information can also be included for node pairs that are not connected by an edge. For example, by counting the number of specific edge labels on all shortest paths between two nodes.

# F  FURTHER APPLICATIONS: IMAGE AND TEXT CLASSIFICATION

Our approach is not limited to the graph domain. Indeed, there are multiple options to extend it to other domains. We will give examples how image and text data can be represented as graphs and how our approach can be applied to them.

**Image Data**   Each image can be seen as a grid graph with diagonal edges. Ordinary convolutional neural networks (CNNs) are based on fixed-size kernels that move along the image. With our novel approach applied to the above interpretation of images as grid graphs, message passing in our invariant-based encoder is nothing more than a convolutional kernel of arbitrary shape and size, see Figure 24 for an illustration.

**Text Data**   Each text can be seen as a path graph where each node corresponds to a word in the text, see Figures 25 and 26 for an illustration. Such representations of text are also called *local word consecutive* (Wang et al., 2024). The label of a node in this case can be the id of the corresponding word given by a tokenizer. This allows to apply our approach directly to text classification by using the path graph representation. Learning the weights of the invariant-based encoder is nothing more than learning relations between certain words at a certain distance in the text. The application of our approach to text classification can be seen as a generalization of the approach of Huang et al. (2019).

Notably, interpretability and transferability of our approach also applies to these domains. Specifically, we can re-use the learned weights between two words in a text to learn relations between two words in another text.

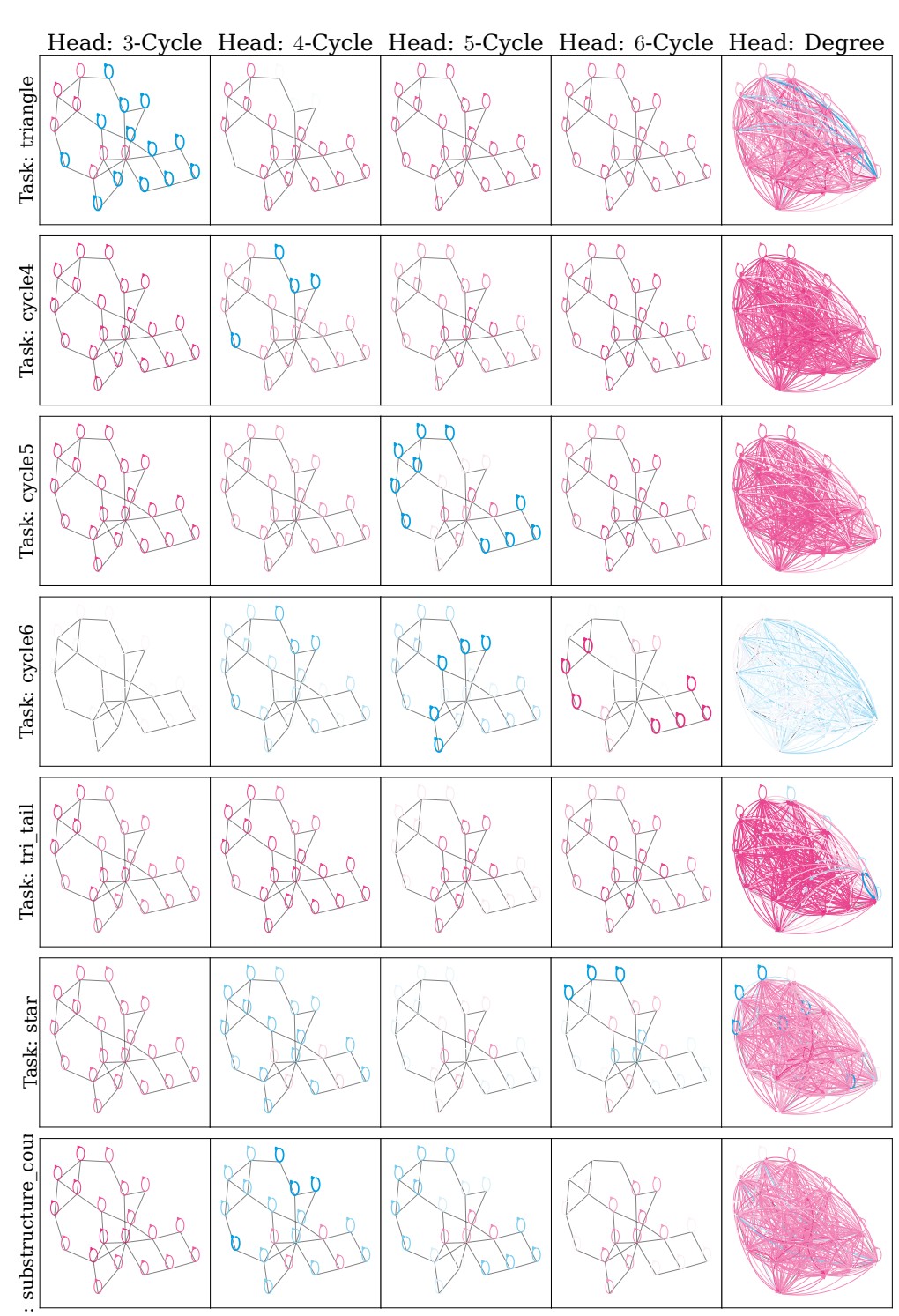

Figure 11: Message passing within different heads of the ShareGNNs for the substructure counting task on one example graph from the test set. The tasks from top to bottom are *triangle, cycle4, cycle5, cycle6, tailed triangle and star*. The last row visualizes the ShareGNN (all) that is trained on all tasks in parallel. Each column corresponds to one head of the invariant based encoder, where the heads base from left to right on the graph invariants *3-cycles, 4-cycles, 5-cycles, 6-cycles and degrees*.

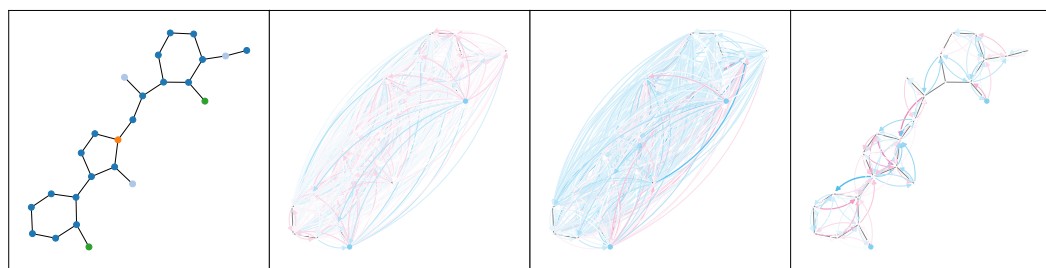

Figure 12: Message passing within different heads of the ShareGNN for the ZINC dataset on one example graph. The leftmost column shows the original graph with atomic number node labels. The other columns show the message passing weights for different heads of the invariant-based encoder. The parameters per head are from left to right: *(1) induced cycles (lengths 6), $d > 1$, (2) atomic numbers, $d > 1$, (3) 1-WL labels at iteration* 1 *(using edge and atomic numbers as initial label),* $d \in \{1, 2\}$.

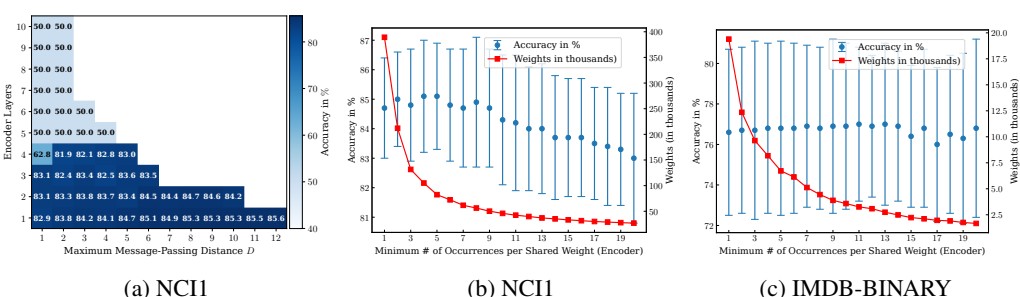

| (a) NCI1 | (b) NCI1 | (c) IMDB-BINARY |

Figure 13: Ablation Study. (13a) ShareGNN performance for different maximum message-passing distances $D$ ($x$-axis) and different numbers of encoder layers ($y$-axis). (13b) and (13c) ShareGNN performance for different number of weights. The model corresponding to the $x$-axis value $i$ contains only those shared message passing weights that occur more than $i - 1$-times in the respective graph dataset.

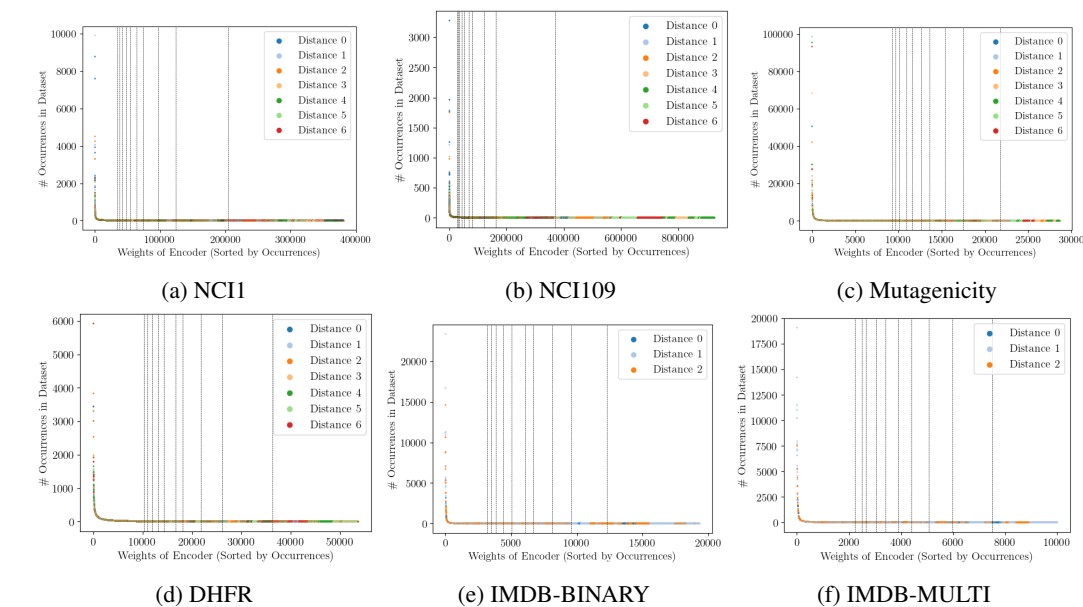

Figure 14: Number of occurrences of the message-passing weights of the encoder layer summed over all graphs for the respective dataset. The model is initialized with the best hyperparameters for the respective dataset. The ten vertical dotted line indicate the distribution of the number of occurrences. All weights in the plots that are right of the $n$-th vertical line (counting from the right) are present less than $(n + 1)$-times in the dataset, i.e., weights left of the leftmost vertical line appear more than 10 times in the dataset. The different colors denote the node pair distance the respective weight is associated with.

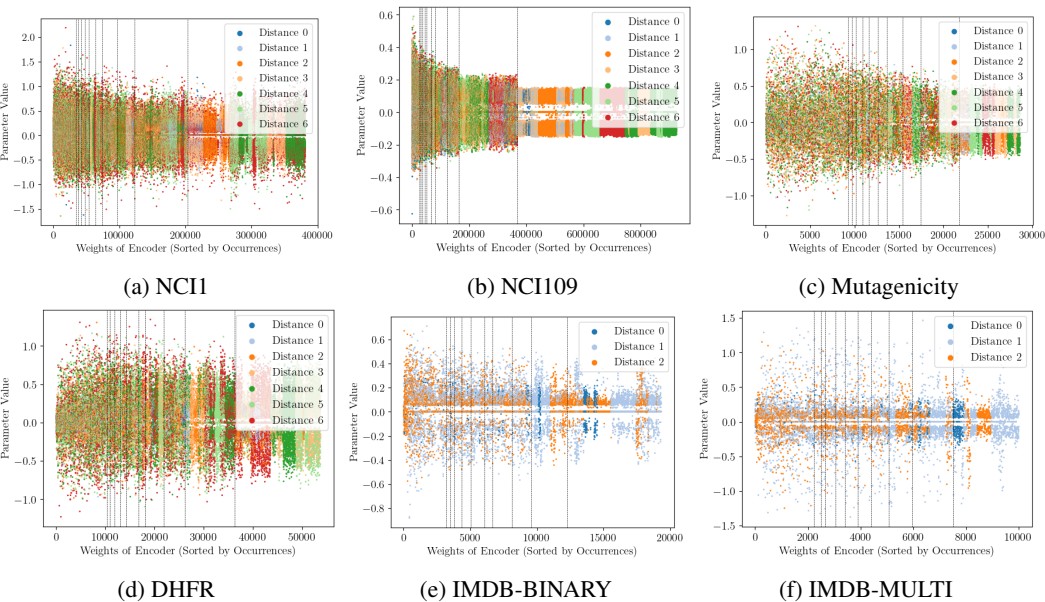

Figure 15: Values of the trained message-passing weights of the respective datasets. The weights are sorted by the number of occurrences in the dataset. The model is initialized with the best hyperparameters for the respective dataset. The ten vertical dotted lines indicate the number of occurrences. All weights in the plots that are right of the $n$-th vertical line (counting from the right) are present less than $(n + 1)$-times in the dataset, i.e., weights left of the leftmost vertical line appear more than 10 times in the dataset. The different colors denote the node pair distance the respective weight is associated with.

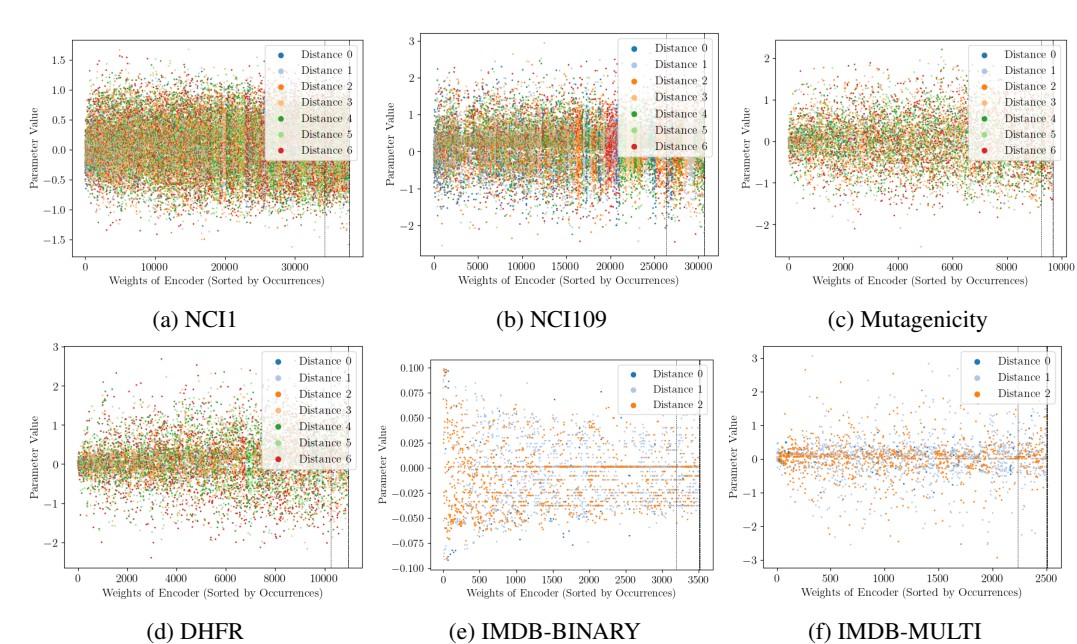

Figure 16: See Figure 15, but this time the model is trained only with message-passing weights that occur at least 10 times in the dataset.

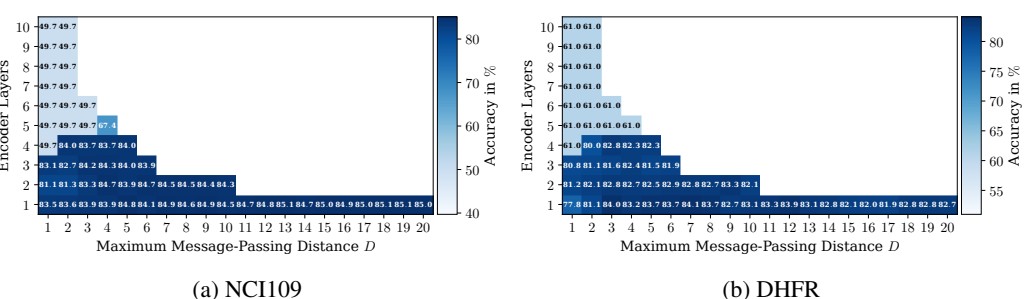

Figure 17: ShareGNN performance for different maximum message-passing distances $D$ ($x$-axis) and different numbers of encoder layers ($y$-axis).

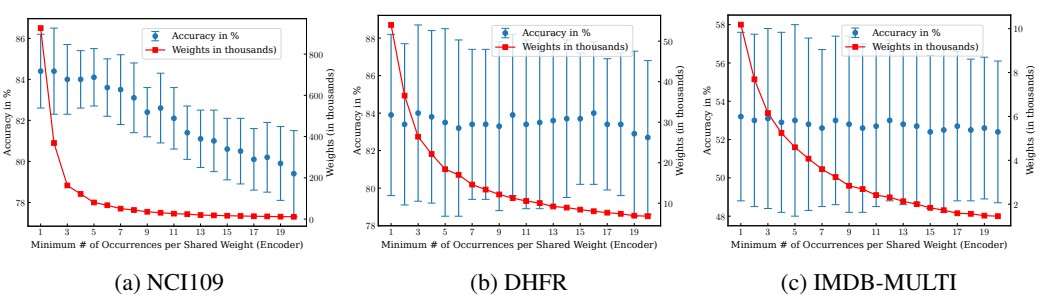

Figure 18: ShareGNN performance (mean accuracy and standard deviation) for different number of weights. The model corresponding to the $x$-axis value $i$ contains only those shared message passing weights that occur more than $i - 1$-times in the respective graph dataset.

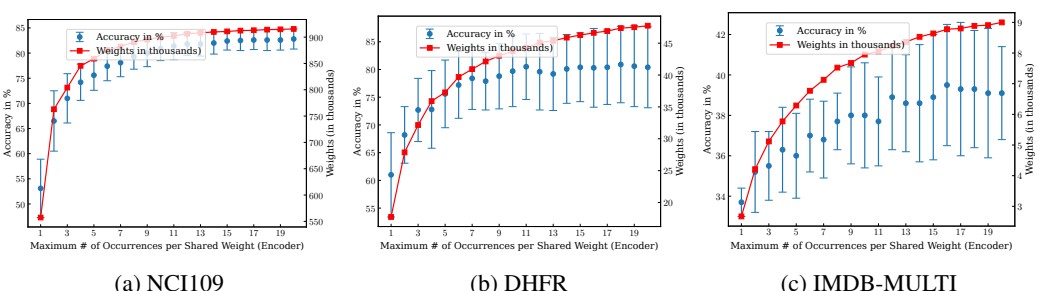

(a) NCI109        (b) DHFR        (c) IMDB-MULTI

Figure 19: ShareGNN performance (mean accuracy and standard deviation) for different number of weights. The model corresponding to the $x$-axis value $i$ contains only those shared message passing weights that occur less than $i - 1$-times in the respective graph dataset.

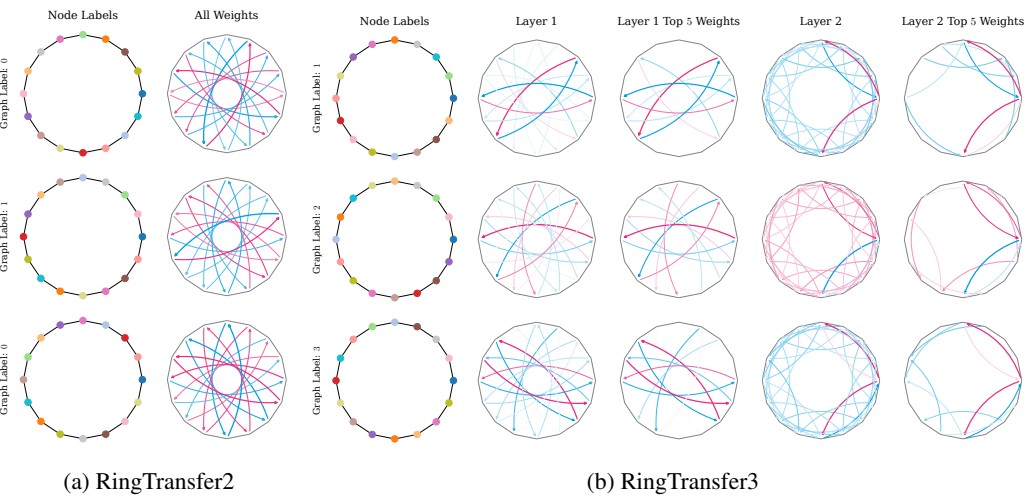

(a) RingTransfer2               (b) RingTransfer3

Figure 20: Visualization of the weights for the respective dataset. The first column of each subfigure shows the graphs with the colors of the nodes representing the different node labels. The other columns show all learned weights and the top five absolute weights, respectively, for the respective encoder layer.

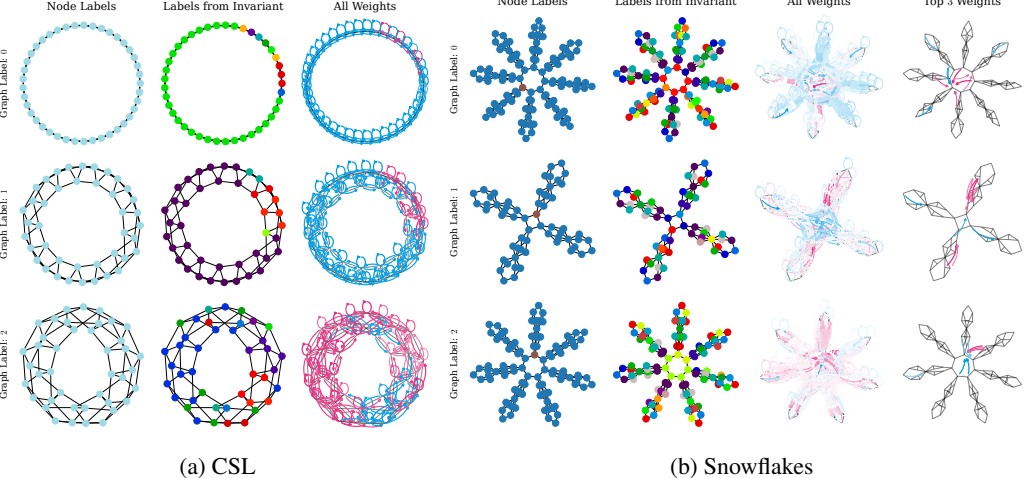

(a) CSL               (b) Snowflakes

Figure 21: Visualization of the weights for the respective datasets. The first column of each subfigure shows the graphs with the colors of the nodes representing the different node labels. The second column shows the labels of the nodes with respect to the labeling function. The last two columns in (21b) show all and the top three absolute weights.

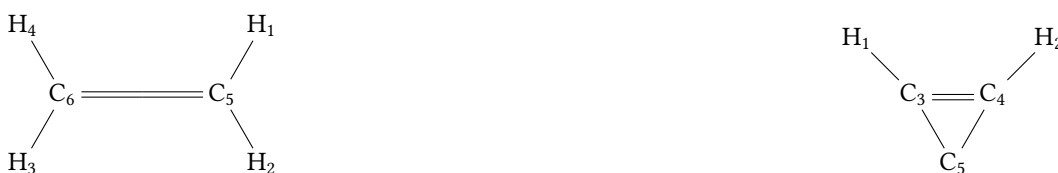

Figure 22: Molecular graphs of ethylene (left) and cyclopropenylidene (right). The numbers denote the order of the nodes.

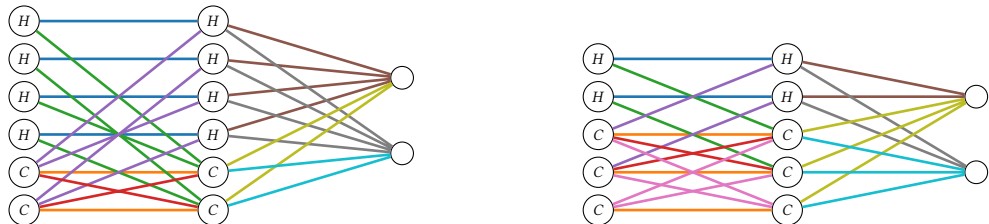

Figure 23: Computational graphs of a simple ShareGNN with one encoder layer for the molecular graphs of ethylene (left) and cyclopropenylidene (right). The input signal is propagated from left to right. The graph nodes represent the neurons of the neural network. Edges of the same color denote layer-wise shared weights.

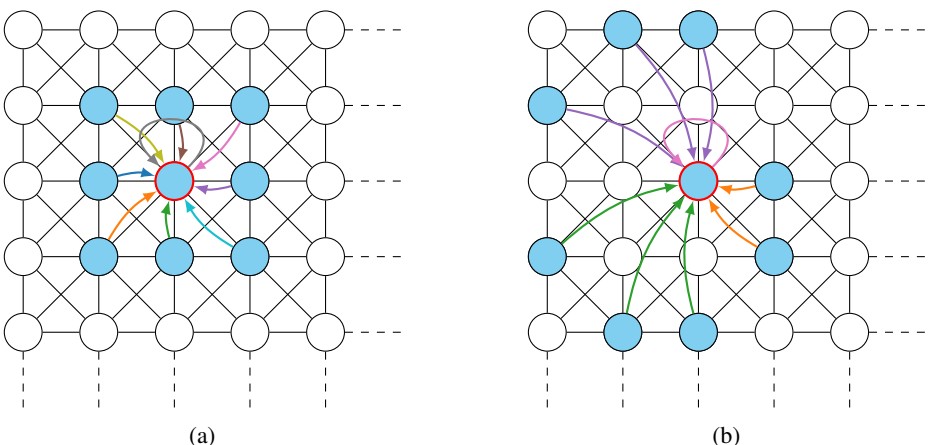

(a)                                          (b)

Figure 24: Representation of a 5x5 partial image as a grid graph with diagonal edges. The left subfigure shows the ordinary convolution for images with a regular kernel (3x3 square shape). The right subfigure shows an example of an irregular kernel (G-shape) that can be represented by our invariant-based encoder layer. The node with the red border corresponds to the center of the kernel, the light blue ones denote the receptive field. Different edge colors denote different (message-passing) weights.

Texts are nothing else than labeled path graphs .

⟩ Tokenization

[CLS] texts are nothing else than labeled path graphs . [SEP]

BERT: 101 6981 2024 2498 2842 2084 12599 4130 19287 1012 102

⟨101⟩—⟨6981⟩—⟨2024⟩—⟨2498⟩—⟨2842⟩—⟨2084⟩—⟨12599⟩—⟨4130⟩—⟨19287⟩—⟨1012⟩—⟨102⟩

Text s are nothing else than lab eled path graph s .

GPT-2: 8206 82 389 10528 25974 621 3498 18449 10644 29681 82 13

⟨8206⟩—⟨82⟩—⟨389⟩—⟨10528⟩—⟨25974⟩—⟨621⟩—⟨3498⟩—⟨18449⟩—⟨10644⟩—⟨29681⟩—⟨82⟩—⟨13⟩

Figure 25: Representation of a sample text as labeled path graph. First, the text is parsed through a tokenizer, that assigns each token a unique identifier. Second, each token is mapped to a node in a path graph. That is, the edges of the graph connect consecutive tokens, forming a path.

Text s are nothing else than lab eled path graph s .

GPT-2:

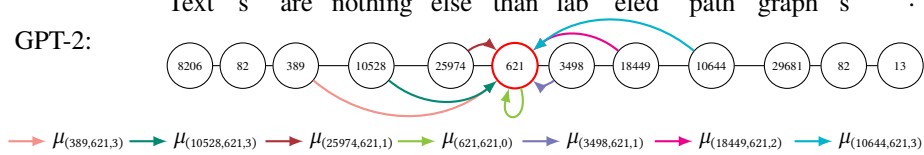

$\longrightarrow \mu_{(389,621,3)}$ $\longrightarrow \mu_{(10528,621,3)}$ $\longrightarrow \mu_{(25974,621,1)}$ $\longrightarrow \mu_{(621,621,0)}$ $\longrightarrow \mu_{(3498,621,1)}$ $\longrightarrow \mu_{(18449,621,2)}$ $\longrightarrow \mu_{(10644,621,3)}$

Figure 26: Encoder Layer: Message passing in the labeled path graph generated by the GPT-2 tokenizer. The arrows indicate different message passing weights $\mu$ from nodes up to distance 3 to the node corresponding to the token *than* (621). Different colors indicate different weights.

