# OpenReview forum: "Weight Sharing for Graph Structured Data"
_ICLR.cc/2026/Conference — Submitted to ICLR 2026_

### Official Review · Reviewer_NgwU · 2025-10-18

**Soundness:** 3
**Presentation:** 3
**Contribution:** 2
**Rating:** 4
**Confidence:** 4

**Summary:**

This paper proposes ShareGNNs, a novel family of permutation-invariant graph neural networks that realize invariant-based weight sharing by indexing weights through graph invariants preserved under node permutations. The approach enables systematic parameter reuse across structurally equivalent subgraphs, offering a principled and efficient mechanism for permutation-aware learning with controllable expressivity. Experimental results on synthetic and real-world benchmarks demonstrate that ShareGNNs achieve competitive performance with minimal message-passing depth, highlighting both the practicality and interpretability of the proposed paradigm.

**Strengths:**

- The writing is good.
- The weight sharing on node/node pair is quite novel.
- The performance on substructure counting is superior.

**Weaknesses:**

- The expressive ability, scalability, and transferability are limited.
- High computation cost on shortest distance computation.
- The performance on real-world datasets is limited.

**Questions:**

* The model design relies on node pairs observed in the training set. Consequently, if crucial node pairs are absent during training, generalization and overall effectiveness may be compromised.
* The computational complexity of computing pairwise distances is underestimated. It is not merely $O(n^2)$, but rather $O(n^2) \times O(\text{SP})$, where $O(\text{SP})$ denotes the complexity of the shortest path algorithm used.
* Report the actual preprocessing time and compare it against the training time to assess overall efficiency.
* In implementation, the model constrains the maximum pairwise distance, which inherently restricts long-range interactions and conflicts with the second stated design objective. Moreover, this truncation can exclude important structural patterns in large graphs, raising concerns about scalability.
* Since all node-pair weights are learned, the model ignores the original edge connectivity of the graph, which may weaken its ability to leverage inherent structural information.
* Although an edge-labeled variant is proposed, it sacrifices the ability to capture long-range interactions, potentially explaining the weak performance on the ZINC dataset.
* The baselines used for substructure counting are outdated. Compare with recent SOTAs, such as Graph as Point Set (ICML 2024).
* Given that node labels already encode 1-WL information, it is unclear why the model underperforms compared to the WL kernel.
* The substructure counting task leverages pattern-label information, which gives this model an advantage, making the comparison with baselines potentially unfair.
* What is the performance on large graphs?
* What is the performance with various model depth?
* A sensitivity analysis on the number of weight-sharing heads is needed.
* Finally, the authors should discuss whether the proposed approach can be extended to node classification and link prediction tasks.

---

> ### Author Response · Authors · 2025-11-21
> **Reply to Reviewer NgwU**
>
> We thank the reviewer for their thoughtful comments! We are happy about the opportunity to clarify some concerns and misunderstandings.
>
> **W1 The expressive ability, scalability, and transferability are limited.**
>
> > **No,** as stated in the paper, the approach can be made arbitrarily expressive by choosing a sufficiently powerful graph-invariant.
> **Please explain your concerns regarding transferability in more detail.**
> By design, ShareGNNs weights can be transferred to graphs of different sizes and structures using the graph-invariants thus
> the concerns regarding transferability are unclear to us.
>
> **W2 Scalability**
>
> > **We disagree** that this is a specific problem of ShareGNNs.
> See our answer to Reviewer LeQ3 regarding scalability.
>
> **W3 Performance**
>
> > **We disagree.** Our main contribution is the novel framework for structure based weight sharing using graph-invariants.
> It can be used in various ways, one of them being ShareGNN.
> Yet, it is strong enough to achieve SOTA performance on real-world and synthetic datasets.
> We are happy to explore further datasets in future work.
>
> **Q1: The model design relies on node pairs [...]**
>
> > **This is not a limitation specific to ShareGNNs**, distributional shifts between train and test set is a general problem in machine learning.
>
> **Q2 The computational complexity of computing pairwise distances is underestimated...**
>
> > **Yes and no.**  We will address this in the revised version.
> For sparse graphs, $m=O(n)$, the complexity reduces to $O(n^2)$.
> In general, the complexity is $O(n \cdot (m+n))$, running $BFS$ ($O(m+n)$) $n$-times, i.e., $O(n^3)$ in the worst case if $m=O(n^2)$.
>
> **Q3 Report the actual preprocessing time [...]**
>
> > **We reported the times in Tables 7 and 8 in the appendix**.
> See also our answer to Reviewer LeQ3.
>
> **Q4 In implementation, the model constrains the maximum pairwise distance, [...]**
>
> > **Constraining the maximum pairwise distance is a choice, not a limitation.**
> We can use **arbitrary pairwise distances (also in the implementation)** if desired.
> However, we found out that the performance does not significantly
> improve when using larger maximum pairwise distances, see Figure 17 in the appendix where we vary the maximum distance almost up to the diameter of the graphs in the dataset.
>
> **Q5 Since all node-pair weights are learned, the model ignores [...]**
> > **No this is not the case**. The original edge connectivity is incorporated via the distance invariant as distance 1 corresponds to direct neighbors.
> Thus, the model incorporates it as one of the structural patterns.
>
> **Q6 Although an edge-labeled variant is proposed, it sacrifices [...]**
> > **No**, long-range interactions can still be captured via the distance invariant even in the edge-labeled variant.
> Regarding the performance on ZINC, it could be that there are other information encoded by the competitors, that we
> do not use in our current hyperparameter configuration.
> It is an interesting direction and for us also a advantage of our approach that we can look which invariant
> patterns are important for the task.
> We will explore this in future work.
>
> **Q7 The baselines used for substructure counting [...]**
> > **We do not agree**, that the baselines are outdated, and we have to decide on some baselines due to space limitations.
> Moreover, we included several recent GNN architectures for the substructure counting task.
>
> **Q8 Given that node labels already encode 1-WL [...]**
> > **This is an open research question in the GNN community** why WL-kernel based methods perform better than 1-WL based GNNs
> on some molecular datasets
> even when message passing GNNs are as expressive as 1-WL.
> We can only speculate that WL-kernel based methods are more robust to overfitting due to their kernel nature.
>
>
> **Q9 The substructure counting task leverages pattern-label information, which gives this model an advantage [...]**
> > **No, the comparison is not unfair.**
> We do not use the counts directly showing that our model is able to learn to count the relevant patterns via the structure based weight sharing.
> Also, competitors encode the relevant pattern information (i.e., circles) in their models, too, hidden in the architecture design.
> Indeed, we see this as an advantage of our approach that we can easily incorporate almost arbitrary structure based information via graph-invariants without designing specific architectures for specific information encodings.
>
> **Q10 - Q12 [...]**
> > As stated above our main focus is to introduce a novel framework.
> Thus, we see these questions as interesting future research directions but not as limitations of our work.
>
> **Q13 Finally, the authors should discuss whether [...]**
> > The approach can be extended to node classification and link prediction tasks as we can
> modify the decoder accordingly to output node-level or edge-level predictions.
> **We can add a short explanation** right after "extending it to node classification, link prediction, [...]" in Section 2.5.

---

> > ### Comment · Reviewer_NgwU · 2025-11-26
> > **Thanks for your rebuttal**
> >
> > W1 & Q1. As you responded, the effectiveness is largely attributed to selecting sufficiently expressive handcrafted graph invariants. This process is highly domain-specific, expert-dependent. For generalizability, conventional GNNs use the message-passing mechanism, which is a more fundamental node feature combination mechanism and ensure the generalizability. In contrast, ShareGNNs are constrained by predefined invariants, which introduce strong inductive bias and thus limit the expressive ability of neural networks.
> >
> > W2, Q2 & Q3. Regarding scalability, Tables 7 and 8 indicate that preprocessing time is already longer than model training time on several datasets. This suggests limited practicality when scaling to large graphs. Additionally, for the response to Reviewer LeQ3, the time comparison with prior work is not meaningful since the results are obtained using different hardware environments.
> >
> > Q4. Figure 17 exactly shows the importance of incorporates longer distance. On NCI1, good performance is achieved when $D \ge 13$ (diameter=11.3 in Table 4).
> >
> > Q6. Yes, the “other information” used in prior work may be the ability to jointly model both node and edge features, which is missing in this work.
> >
> > Q7. ICLR allows an unlimited appendix, so the justification of space constraints is not sufficient.
> >
> > Q9. So, I may wandering what is the performance without the pattern-label information in substructure counting task.
> >
> > Q10–Q12. As you stated, the main focus is to introduce a novel framework, thus, it is better to help readers have a comprehensive understanding of this framework, not just from several fixed angles.

---

> > > ### Author Response · Authors · 2025-12-04
> > > **Answer Reviewer Comment**
> > >
> > > We thank the reviewers for their answer and would like to clarify the following questions:
> > >
> > > W1 & Q1. As you responded, the effectiveness is largely attributed to selecting sufficiently expressive handcrafted graph invariants. This process is highly domain-specific, expert-dependent. For generalizability, conventional GNNs use the message-passing mechanism, which is a more fundamental node feature combination mechanism and ensure the generalizability. In contrast, ShareGNNs are constrained by predefined invariants, which introduce strong inductive bias and thus limit the expressive ability of neural networks.
> > >
> > > > **No, as stated in the paper, the approach can be made arbitrarily expressive by choosing more expressive graph-invariants. Thus, it is not limited in its expressive ability.**
> > > > Moreover, the experiments show that using the invariants already leads to improved performance compared to classical GNNs on various tasks. Thus, we do not see this as a limitation of our approach.
> > > > In our approach node features can still be used in addition to the graph-invariants thus allowing to combine both sources of information.
> > >
> > >
> > > W2, Q2 & Q3. Regarding scalability, Tables 7 and 8 indicate that preprocessing time is already longer than model training time on several datasets. This suggests limited practicality when scaling to large graphs. Additionally, for the response to Reviewer LeQ3, the time comparison with prior work is not meaningful since the results are obtained using different hardware environments.
> > >
> > > > **No, preprocessing is only done once and thus not a bottleneck also for large datasets compared to training time.**
> > > > Moreover, regarding the answer to Reviewer LeQ3, we stated our hardware environment in the appendix which is a small/standard CPU computer (AMD Ryzen 9 7950X 16-core processor with
> > > 128 GB of RAM) compared to the prior work:
> > > >> All experiments were conducted on a cluster with 12 NVIDIA A10 GPUs (24 GB) and 4 NVIDIA H100s. Each node had 64 cores of Intel(R) Xeon(R) Gold 6326 CPU at 2.90GHz and 500GB of RAM. All experiments used at most 1 GPU at a time.
> > >
> > >
> > >
> > > Q7. ICLR allows an unlimited appendix, so the justification of space constraints is not sufficient.
> > >
> > > > This is correct, but we do not have unlimited time to implement and run all possible baselines.

---

### Official Review · Reviewer_LeQ3 · 2025-10-28

**Soundness:** 2
**Presentation:** 3
**Contribution:** 3
**Rating:** 4
**Confidence:** 3

**Summary:**

This paper introduces "invariant-based weight sharing," a novel paradigm for graph neural networks where learnable weights are indexed directly by graph invariants (e.g., node labels, pairwise distances) rather than being fixed or computed from node features. This approach is designed to enable principled parameter reuse across structurally equivalent regions. The authors instantiate this concept in a new model, ShareGNN, which uses an encoder-decoder architecture to achieve global, transformer-like message passing in a single layer. The paper provides theoretical analysis linking the model's expressivity to the discriminative power of the chosen invariants and reports competitive experimental results on subgraph counting, synthetic benchmarks, and real-world graph classification and regression tasks, highlighting the efficacy of its shallow architecture.

**Strengths:**

1. The core idea of "invariant-based weight sharing" presents a interesting paradigm for parameterizing graph neural networks.

2. The proposed ShareGNN model offers a potential solution to the over-smoothing issues of deep GNNs.

3. The paper supports its claims with a comprehensive evaluation, including theoretical analysis of expressivity and empirical results across a diverse set of benchmarks.

**Weaknesses:**

1. The method is not scalable. It relies on $O(n^3)$ preprocessing (all-pairs shortest paths) and an $O(n^2)$ layer computation. These costs are dismissed as "negligible" only because the experiments are confined to extremely small graphs (e.g., avg. 30 nodes). It lacks any experiments on medium or large benchmarks (like OGB).

2. The core mechanism appears to be a combination of existing ideas: it uses R-GCN-style relational indexing (where the "relation" is a tuple of invariants) to parameterize a Graph Transformer-style global message passing layer.

3. The main theoretical results (Propositions 1, 2, and 3) merely prove permutation equivariance and invariance. This is a fundamental design requirement for any GNN and not a novel contribution of this specific architecture.

**Questions:**

The ablation studies in Appendix D.5 (e.g., Figures 13 & 18) are very insightful. They show that over 80% of the model's parameters—specifically, the weights corresponding to infrequent structural invariants—can be pruned with almost no loss in performance. Does this not strongly suggest that the proposed parameterization scheme is massively over-parameterized by design, and that the vast majority of learned weights are ultimately unnecessary for the task?

---

> ### Author Response · Authors · 2025-11-21
> **Reply to Reviewer LeQ3**
>
> We thank the reviewer for their comments and the opportunity to clarify our work.
> Below we address the main concerns one by one.
>
> **W1: The method is not scalable. [...]**
>
> > **We disagree.** Our method is comparable or better in terms of runtime complexity to recent graph learning preprocessing steps
> (e.g., computing homomorphism counts [1], subgraph isomorphism counts [2], computing topological features [3] etc.) or to common global message passing layers (e.g., transformer-based GNNs).
> By definition "Graph Transformer-style global message passing layer" have $O(n^2)$ complexity per layer.
> We do not see why especially our method should be singled out in this regard.
>
> > Moreover, as also mentioned in the paper, we employ various techniques (e.g., restricting the maximum distance) to improve scalability.
> Table 7 shows that the preprocessing time of the pairwise distances as well as the time to preprocess the node labels is not a bottleneck in practice; in fact,
> **we are faster than recent papers using similar preprocessing steps**, see e.g. [1] 360s for ZINC 12-k compared to ours 130s.
>
> > Finally, we would like to point out that scalability is not the main focus of our work.
> Our main focus is to introduce a novel framework for structure based weight sharing in GNNs using graph-invariants
> and to show that this approach leads to improved performance and interpretability on various graph learning tasks.
> We believe that scalability can be improved in future work by employing various techniques (e.g., sampling, hierarchical methods, sparse approximations etc.) without changing the core idea of our approach.
>
> **W2 The core mechanism appears to be a combination of existing ideas: [...]**
> > **Not quite.** We thank the reviewer for pointing out R-GCN as related work and we will discuss it in the related work section of the revised paper.
> > However, we would like to clarify that **R-GCN differs from ShareGNN in several important aspects**:
>
> > - R-GCN does not consider structure based invariants and hence also does not use structure based weight sharing as we do in ShareGNN.
>
> > - The approach is based on learnable normalizing factors while we directly consider learnable relations (based on graph-invariants).
>
> > - Only node classification and link prediction tasks are considered as their weight matrices are of fixed size in contrast to our dynamic sized weight matrices (see also the answer to Reviewer fZtg).
> One of the main practical challenges (on the implementation side) and also one of the main novelties in our approach is the construction of such dynamic sized weight matrices.
>
> > - Besides an encoder, we also provide a decoder to handle graph-level tasks.
>
>
> **W3 The main theoretical results (Propositions 1, 2, and 3) merely prove permutation equivariance and invariance. [...]**
> > **No**, while all GNNs are permutation invariant/equivariant, their permutation awareness is limited to the local neighborhood aggregation scheme.
> In contrast, Propositions 1, 2, and 3 show that ShareGNN's use of graph-invariants allows it to be permutation aware at a global level **beyond** local neighborhoods.
> The only way to achieve both, global permutation awareness and message passing via arbitrary long-range interactions is to use graph-invariants as we do in ShareGNN.
>
> **Questions:[...] Does this not strongly suggest that the proposed parameterization scheme is massively over-parameterized by design,
> and that the vast majority of learned weights are ultimately unnecessary for the task?**
>
> > **Yes, but this was not the point of the study.**
> Your observation is correct as it is for most neural networks (think the Lottery Ticket Hypothesis [4]).
> This ablation study has a different goal: not only do we show that for ShareGNNs a large fraction of the weights can be removed without significant performance degradation,
> but hint at the inherent interpretability of the remaining weights.
> Detailed analyses may provide insights into which structural patterns are important for a given task.
> We leave the further study of the ShareGNN's proper interpretability for future work.
>
>
> [1] Bao et al. Homomorphism counts as structural encodings for graph learning. 2025.
> [2] Bouritsas et al. Improving graph neural network expressivity via subgraph isomorphism counting. 2023.
> [3] Bodnar et al. Weisfeiler and Lehman Go Cellular: CW Networks. 2021.
> [4] Frankle and Carbin. The Lottery Ticket Hypothesis: Finding Sparse, Trainable Neural Networks. 2019.

---

### Official Review · Reviewer_W22x · 2025-10-29

**Soundness:** 3
**Presentation:** 3
**Contribution:** 2
**Rating:** 4
**Confidence:** 3

**Summary:**

This paper proposes a novel Graph Neural Network (GNN) architecture called ShareGNN, whose core idea is to index parameter weights through graph invariants, enabling sharing and generalization across node pairs. The paper demonstrates a certain degree of novelty.

**Strengths:**

(1) Introducing graph invariants as a key to parameter sharing is an interesting viewpoint.
(2) The concept is clearly explained and well-organized, logically connecting the core idea to a new architecture (ShareGNN) and theoretical proofs.

**Weaknesses:**

(1) Although the invariant-based formulation is conceptually interesting, its empirical validation remains modest. Most benchmarks are standard small- to mid-scale datasets. Large-scale or high-density graphs (where O(n²) cost dominates) are not convincingly analyzed.
(2) The authors claim comparable asymptotic complexity to Graph Transformers but with better parameter efficiency. However, no direct runtime or memory comparison with Graph Transformers is reported. In practice, the O(n²) operations for all node pairs could still be prohibitive for large graphs, unless sparsity or truncation effects are explicitly quantified.
(3) There is limited exploration of how invariant choices (e.g., WL depth, pattern size, distance) affect performance and cost. It remains unclear which invariants are most influential and whether combinations always yield improvements.

**Questions:**

See in Weaknesses.

---

> ### Author Response · Authors · 2025-11-21
> **Reply to Reviewer W22x**
>
> We thank the reviewer for their comments and the opportunity to clarify our work. We are excited to hear that the reviewer finds our invariant-based formulation conceptually interesting!
>
> Below we address the main concerns one by one.
>
> **(1) Although the invariant-based formulation is conceptually interesting, its empirical validation remains modest. [...]**
>
> > **We disagree with the comment's premise.** Our main contribution is the introduction of a novel framework for structure based weight sharing in GNNs using graph-invariants.
> This framework is general and can be used in various ways, one of them being ShareGNN.
> Yet, it is strong enough to achieve SOTA performance on 5 out of 7 real-world datasets as well as on the synthetic datasets designed to test specific graph learning capabilities.
> Further, our approach offers interpretable insights into what the models has learned on the various graph learning tasks (see the ablation studies).
> While there is still improvement for some specific tasks and datasets, we believe that it is not the goal of a single paper to solve all problems but rather to introduce new ideas and frameworks that can be built upon in future work.
> Thus, we leave the improvement of performance on specific tasks and datasets for future work.
>
> **(2) The authors claim comparable asymptotic complexity to Graph Transformers [...]**
>
> > In theory both ShareGNN and Graph Transformers have $O(n^2)$ complexity per layer due to considering pairwise relations.
> If desired, we can include a memory and runtime comparison in the revised paper.
>
> **(3) There is limited exploration of how invariant choices (e.g., WL depth, pattern size, distance) affect performance.**
>
> > **No**, our ablation studies include results on different invariant choices. Regarding different distances, Figure 17 in the appendix shows that performance does not significantly improve but stays stable when increasing the maximum distance.
> Regarding different pattern sizes and WL depths, we only reported the results of the hyperparameter configurations that led to the best performance on the validation set.
> If desired, we can provide more detailed results (performance vs. pattern size and WL depth) in the revised paper or the appendix.

---

### Official Review · Reviewer_fZtg · 2025-10-29

**Soundness:** 2
**Presentation:** 3
**Contribution:** 1
**Rating:** 4
**Confidence:** 4

**Summary:**

The paper introduces ShareGNN, a GNN architecture that leverages parameter sharing based on node labels, substructure patterns, etc’. The authors provide some  theoretical results on permutation invariance and expressiveness and evaluate the method on several graph benchmarks.

**Strengths:**

- The paper is clearly written and easy to follow, with well-structured explanations of the model and its motivation.
-  The proposed approach is simple and conceptually intuitive, making it straightforward to implement and analyze.

**Weaknesses:**

While the idea of exploring new forms of parameter sharing in GNNs is interesting, I find that the paper does not convincingly demonstrate meaningful advantages over existing architectures, either theoretically or empirically. The authores state five main advantages of ShareGNN:  Adaptivity, permutation awareness, expressiveness, Long-range interactions and Transferability and interpretability.  I have some concerns with regards to each of these.

- Adaptivity: The authors highlight that ShareGNN dynamically adapts to graphs of varying sizes. However, this property is common to almost all GNNs, and it is not clear in what sense ShareGNN is more adaptive. In fact, by assigning parameters to multiple label or pattern types, the model may introduce more parameters per layer than a standard MPNN.

- Permutation awareness: This is a standard property of most GNN architectures, so it does not appear to be a distinguishing feature.

- Expressiveness: The expressivity discussion is unconvincing. The paper essentially shows that if one provides ShareGNN with expressive functions as input, it can compute them — which is true but not particularly insightful. There is no clear theoretical or empirical evidence of meaningfull enhanced expressivity beyond the expressivity of the functions used to define the parameter sharing shceme.

- Long-range interactions: The claim that ShareGNN can propagate information across arbitrary node pairs in a single layer applies equally to most transformer-based GNNs. The paper does not clearly establish how ShareGNN achieves this more effectively.

- Transferability and interpretability: This is the one claimed advantage that could be interesting, but the paper does not provide experiments to substantiate it. Demonstrations on transfer tasks or interpretability analyses would have strengthened this point.
Additionally, the experimental section is relatively weak. the aouthors state “standardized evaluation protocols are often missing in graph learning” , but in fact, widely used benchmarks such as ZINC and OGB provide well-defined splits and evaluation pipelines. The paper also uses smaller, less reliable benchmarks and compares primarily to outdated baselines and the performance of ShareGNN on them is modest.

Finally, the novelty claim is somewhat overstated. Parameter sharing in GNNs has been directly explored in prior works such as in [1,2]. Additionally, standard MPNN variants like GIN and GCN, can be thought of as “parameter sharing for GNNS” as they already employ shared weights across nodes. In some ShareGNN configurations (e.g., with WL labels), the model may even have more parameters than a typical MPNN without clear gains in expressivity.


[1] Maron et al’. Invariant and equivariant graph networks. 2018.

[2]  Maron et al’.  Provably powerful graph networks. 2019.

**Questions:**

- Include stronger experimental results — for example, on more modern OGB benchmarks — and provide comparisons to relevant baselines, especially recent substructure-counting GNNs (e.g., [3,4]) as well as imroved results on Zinc.

- Provide clearer theoretical and emperical results that demonstrate a non-trivial improvement in expressivity, beyond the ability to compute the pre-defined functions used by ShareGNN.

- Discuss related weight-sharing architectures in greater depth and position ShareGNN more precisely within this context.
Include interpretability or transfer experiments to support the claimed advantages.

[3] Bao et al’. Homomorphism counts as structural encodings for graph learning. 2024.

[4] Bouritsas et al’. Improving graph neural network expressivity via subgraph isomorphism counting. 2024.

**Details Of Ethics Concerns:**

No concerns

---

> ### Author Response · Authors · 2025-11-21
> **Reply to Reviewer fZtg**
>
> We thank the reviewer for their detailed evaluation and constructive feedback.
> We address the main concerns below.
> However, we think that most of the reviewer's concerns stem from a central misunderstanding.
> We would like to start discussing the following comment, which will bring us to the heart of the problem:
>
> ```"Additionally, standard MPNN variants like GIN and GCN, can be thought of as “parameter sharing for GNNS” as they already employ shared weights across nodes."```
>
> >**This is correct, but orthogonal to our work**.
> ShareGNNs employ **structure aware weight sharing**.
> We try to clarify this misunderstanding by a comparison between GCN and ShareGNN forward passes (which we will also add to the revised paper):
>
> GCN:$$\quad\quad\underbrace{\mathbf{D}^{-\frac{1}{2}}\mathbf{\tilde{A}}\mathbf{D}^{-\frac{1}{2}}}_{n\times n}
> \cdot
> \underbrace{\mathbf{X}^{(h)}}\_{n\times k}
> \cdot
> \underbrace{\mathbf{\Theta}}\_{k\times k}
> $$[1]
>
> ShareGNN:$$\quad\quad\underbrace{\mathbf{W}^G}_{n\times~n}
> \cdot
> \underbrace{\mathbf{X}^{ (h) } }\_{n\times k}
> \cdot
> \underbrace{[\mathbf{\Theta}]}\_{k\times k (\text{optional})}$$(Eq.(3) in the paper)
>
> >Indeed, MPNNs use shared weights across nodes, i.e., the same fixed size ($k\times k$) weight matrix $\mathbf{\Theta}$ is used to transform the features of all nodes.
> However, this weight sharing **does not take into account structural properties of the graph**.
> In contrast, our contribution is **structure aware weight sharing** **instead of**/**on top of** feature based weight sharing.
> More precisely, the weight matrix $\mathbf{W}^G$ depending on the input graph $G$, is constructed using graph-invariants and is of **variable size** $n\times n$
> (which is novel to the best of our knowledge).
> Alternatively, it can be seen as learnable adjacency matrix (adjacency beyond edges) illustrated by Figures 3,11,12 or as drop-in replacement of the normalized adjacency matrix in GCNs.
> Table 12 in the appendix shows that encoding extra structural information via graph-invariants in
> $\mathbf{W}^G$ is superior to using feature based weight sharing (i.e., using only $\mathbf{\Theta}$ as in GCN along with encoding the structural information in the node features).
> Thus, **we do not agree** with the criticism that "Finally, the novelty claim is somewhat overstated. ..."
>
> **Adaptivity**
> >Our claim **is different** from that criticized by the reviewer.
> We state "The model is not constrained to fixed-size weight matrices; it dynamically
> adapts to graphs of varying sizes."
> The novelty is the use of weight matrices $\mathbf{W}^G$ of variable size.
> Our contribution differs fundamentally from feature based weight sharing as used in GCN and, e.g.,
> allows to re-use the **same weight matrix** for graphs of **different node feature dimensions $k$** which is
> impossible in GCNs with fixed sized weight matrices.
>
> **Permutation Awareness**
> >Indeed, most GNNs are permutation invariant/equivariant.
> However, their permutation awareness is limited to the local neighborhood aggregation scheme.
> **In contrast**, ShareGNN's are permutation aware at a global level beyond local neighborhoods different from other GNNs.
> One way to achieve both, global permutation awareness and message passing via arbitrary long-range interactions
> is to use graph-invariants as we do in ShareGNN.
>
> **Expressiveness**
> >We see the incorporation of arbitrary graph-invariants as a strength of our approach
> as no special architecture design is necessary to increase expressiveness.
> Expressiveness beyond the given labels is not our main focus, however, Figure 2 shows our approach distinguishes non-isomorphic graphs that 1-WL cannot distinguish based on the distance information alone.
>
> **Long-Range Interactions**
> >**We note in the paper** that transformer-based GNNs are able to capture long-range interactions.
> However, ShareGNN does this in a message passing manner using graph-invariants to guide the message passing.
> This is different from transformer-based GNNs allowing structure based weight sharing.
>
> **Transferability and Interpretability**
> >We thank the reviewer for pointing out these interesting aspects.
> We agree that these are interesting directions, but we see them as future research directions
> and not as limitations of our work.
>
> **Q1: "Include stronger experimental results ..."**
> >See answer to Reviewer NgwU. Reference "[4]" **is already** included in our experimental evaluation (see GSN in Table 3, Table 11) and we will include "[3]" in our experimental evaluation of the revised paper.
>
> **Q2: "Provide clearer theoretical and empirical results that demonstrate [...] "**
> >See Expressiveness answer.
>
> **Q3: "Discuss related weight sharing [...]"**
> >See our clarification above regarding the misunderstanding about weight sharing in GCNs vs ShareGNN. We will add this clarification to the related work section in the revised paper.
>
> [1] Kipf, T. N., & Welling, M.(2017). Semi-supervised classification with graph convolutional networks. In ICLR 2017.

---

> ### Comment · Reviewer_fZtg · 2025-11-25
> **Response (first part)**
>
> I thank the authors for their detailed response, which helped clarify some important points regarding the paper. Before going through the authors’ comments one by one, I would like to highlight the core issue that still prevents me from recommending acceptance:
>
> **Main Concern**
>
> I still feel that the current manuscript does not provide sufficiently convincing evidence of clear, concrete advantages of ShareGNN over existing GNN architectures. At present, it remains unclear why and when ShareGNN should be preferred over commonly used GNNs, either theoretically or empirically.
> The paper highlights several potential advantages—Adaptivity, Expressiveness, Permutation Awareness, Long-Range Interactions, Transferability and Interpretability, and Empirical Evaluations—but none seem convincing enough in their current form.
>
> **Regarding Adaptivity**
>
> I appreciate the clarification provided in the rebuttal. This property seems promising. However
>
> - The paper does not clearly motivate when this property is helpful; and
>
> - No experiment demonstrates a scenario where adaptivity actually provides a benefit (e.g., transfer learning across graphs with different feature dimensions).
>
> Furthermore, the phrasing “adaptivity: the model dynamically adapts to graphs of varying sizes” does not clearly communicate that this refers to varying node feature dimensionality, not just graph size. This should be made explicit.
> A concrete real-world example demonstrating when this property is helpfull would significantly strengthen the contribution.
>
> **Regarding Permutation Awareness**
>
> The phrase “permutation aware at a global level” remains unclear. In my view, a model is either permutation invariant or it is not; it is not clear how “global awareness” relates to permutation invariance. It seems instead like you are referring to long-range interactions, which is a different concept. This ambiguity should be resolved.
>
> **Regarding Expressiveness**
>
> The paper gives expressivity results, but as I stated in my first review, these results only show that ShareGNN can recover graph invariants that are already provided by the practitioner constructing the model. This is, in my view, a minimal expressivity guarantee, not a strong or surprising result.
> To make these expressivity claims meaningful, I would like to see one of the following:
>
> - Evidence that ShareGNN can compute something beyond the invariants used in its construction, or
>
> - Articulation and demonstration of other useful properties (e.g., strong generalization, improved complexity, robustness), which are often the more relevant reasons why expressive GNNs are interesting in practice.
>
> Expressivity alone is rarely sufficient today unless it clearly leads to improved performance or usability. The authors state in their response that “expressiveness beyond the given labels is not our main focus,” which is fine, but this makes the expressiveness properties of ShareGNN less of a clear benefit.
>
> **Regarding Long-Range Interactions**
>
> The rebuttal acknowledges that other mechanisms used in GNNs, such as attention in graph transformers, already support long-range interactions, but argues that ShareGNN enables long-range interactions via message passing. However, no justification is given as to why such interactions would be better than mechanisms already used for long-range interaction.
> A theoretical or empirical clarification would be valuable. As it stands, this appears as a speculative benefit or a suggestion for future work. Without clearer evidence, these cannot be counted as strengths of the method.
>
> **Regarding Transferability and Interpretability**
>
> The rebuttal states that this is an interesting direction for future work, which again makes it a speculative benefit. Providing experiments showing these properties would strengthen the paper. If not, it is reasonable to leave this for future work, but then stronger evidence must be provided elsewhere.

---

> ### Comment · Reviewer_fZtg · 2025-11-25
> **Response (second part)**
>
> **Regarding Experimental Evaluation**
>
> The rebuttal states that Table 12 shows “superiority” of encoding extra structural information via graph invariants, as opposed to feature-based weight sharing. However, the table offers limited empirical support, and “superiority” is interpreted merely as slightly higher scores.
> If the main message of the paper is that ShareGNN is generally superior, then stronger evidence is needed, such as:
>
> - evaluations on more reliable modern benchmarks such as ZINC or OGB datasets,
>
> - broader comparisons across state-of-the-art GNNs.
>
> As I noted earlier, the current benchmarks are small and less reliable, so stronger empirical grounding is essential if the claim is meant to be generic. I acknowledge that GSN is included, but the empirical section still feels underpowered relative to the strength of the claims being made. Including broader and more modern benchmarks would considerably improve the work.
>
> **Response to Comparison of ShareGNN with MPNNs and Parameter Sharing**
>
> My point is that you phrase your introduction in the following way: images have CNNs so “How can weight sharing be defined for irregular structures such as graphs?”. This question already has simpler answers, mainly: MPNNs and also other weight sharing GNNs mentioned in my first review. As such, this nerative slightly overstates the novelty of ShareGNN. I do however agree with the authors that structure aware weight sharing is new, it’s just not the first answer. To the stated question. I thus believe the flow of the introduction should be changed to reflect this.
>
> **Final Position**
>
> If the authors cannot formally define and empirically validate clear advantages of ShareGNN over existing GNNs (which may be difficult within the rebuttal timeline), then I will be keeping my current score. I believe the paper contains promising ideas, but as it stands, the contributions are not sufficiently supported to justify acceptance at ICLR.

---

> ### Author Response · Authors · 2025-12-04
> **Answer to Response (first part/second part)**
>
> We thank the reviewer for their **very detailed answers** and efforts to help us improve our work.
> We are glad
> that we could clarify the misunderstanding regarding the weight sharing mechanism of ShareGNN
> "I appreciate the clarification provided in the rebuttal. This property seems promising. However [...]"
>
> **Regarding Adaptivity**
>
> > **We do not agree with the remaining concerns:**
> >> "The paper does not clearly motivate when this property is helpful; and"
> >> "No experiment demonstrates a scenario where adaptivity actually provides a benefit (e.g., transfer learning across graphs with different feature dimensions)."
>
> > We motivate the usefulness mentioning several scenarios, i.e., transfer learning, interpretability, and application to different domains.
> as well as providing evidence in the experiments stating that:
> >> "Invariant-based weight sharing makes it unnecessary to encode node
> attributes explicitly. In these experiments, the initial node features are set to a constant scalar (1),
> and all structural information enters the model solely through the labeling functions."
>
> > **This shows that ShareGNN can learn from structure only without node features while the competing GNNs need node features to achieve good performance.**
>
> **Regarding Permutation Awareness**
>
> > **As stated above** classical GNNs are permutation invariant/equivariant but only at a local neighborhood level.
> In contrast, ShareGNN's use of graph-invariants allows it to be permutation aware and at the same time
> perform message passing via arbitrary long-range interactions.
> This is different from other GNNs and is an important property of ShareGNN.
>
> **Regarding Expressiveness**
> > **As stated above** it is not our main focus to improve expressiveness beyond the given labels.
> Figure 2 shows an improved expressiveness of ShareGNN based on distance information alone.
> However, this is ignored by the reviewer.
>
> **Regarding Long-Range Interactions and Transferability/Interpretability and Experimental Evaluation**
> > **As stated above** our main focus is on the structure based weight sharing mechanism not on showing that ShareGNN outperforms all other GNNs.
> Thus, we see these interesting aspects as future research directions and not as limitations of our work.

---

### Author Response · Authors · 2025-11-21
**Rebuttal Summary**

We thank all the reviewers for their detailed evaluation and constructive feedback. First and foremost, we are happy to hear that overall, our framework for structure based weight sharing in GNNs using graph-invariants is perceived as an **interesting and novel** weight sharing paradigm (W22x, LeQ3, NgwU) that is **simple and intuitive** (fZtg). Further, we are content that not only our empirical results (LeQ3) and our superior substructure counting (NgwU), but also our theoretical analyses and proofs were appreciated (W22x, LeQ3).

Below, we want to summarize our answers to common questions and re-occurring misconceptions. We are thankful for the feedback and will use it to polish and rephrase relevant sections in an updated version of the paper.

**(1) Novelty**

Three out of four reviewers highlighted how interesting our weight sharing paradigm based on graph-invariants is.
Reviewer fZtg's concern regarding the novelty of our contribution seems to stem from a central misunderstanding: We agree that GIN, GCN and other MPNN variants use shared weights across nodes. However, this weight sharing **does not take into account structural properties of the graph**. In contrast, our contribution is **structure aware weight sharing** **instead of**/**on top of** feature based weight sharing. In our answer to Reviewer fZtg, we clarify this misunderstanding comparing the GCN and ShareGNN forward passes (which we will also add to the revised paper).
Our experiments show (Table 12 in the appendix together with Table 2) that our information encoding is superior to node feature based weight sharing alone.


**(2) Scalability**

We agree that scalability is an important theoretical aspect of graph learning methods.
We are therefore happy to explain that our methods offers competitive runtime compared to state-of-the-art GNN architectures:
our preprocessing computing pairwise distances is comparable or less expensive than popular preprocessing steps (e.g., computing homomorphism counts, subgraph isomorphism counts, computing topological features etc.)
and the complexity considering pairwise relations is by definition $O(n^2)$ per layer, just as it is the case for transformer-based GNNs.
Thus, it is **not right** to single out our method in this regard.
In particular, when solving problems such as substructure counting (which is NP-hard in theory) with GNNs it is unreasonable to expect that the proposed model has quadratic or better runtime complexity.

**(3) Experiments**

Our paper's scientific contribution is the introduction of a novel framework for **structure based weight sharing in GNNs** using graph-invariants.
This mechanism can be used in various ways as proposed in the paper, only one of them being ShareGNN.
Yet, our experiments show that our ShareGNN achieves state-of-the-art performance on 5 out of 7 real-world datasets as well as on all synthetic datasets designed to test specific graph learning capabilities.
We agree that there is still room for improvement for some specific tasks and datasets.
However, we leave the improvement of performance by hyperparameter tuning for specific tasks and datasets for future work.

---

### Meta-Review · Area_Chair_X9gR · 2025-12-29

**Summary:**

In this paper, the authors study weight sharing for graph machine learning by focusing on graph invariants, which enable permutation-aware learning. Based on this principle, the authors propose ShareGNNs using an encoder-decoder design. Theoretical guarantees and experimental verifications are provided for the proposed method.

Though reviewers acknowledge that the paper provides an interesting perspective, several critical concerns were raised regarding insufficient experiments, computational overheads, as well as the advantages of the proposed method over existing methods. Overall, the paper provides some interesting findings that could be explored further, but in its current form, as reflected by the unanimous negative initial scores and not responding positively to the rebuttal (for two participating reviewers), it is quite clear that the paper does not meet the bar for acceptance. Thus, I recommend rejection.

**Reviewer Concerns:**

Reviewer fZtg:
Most critical weakness-convincing evidence of clear, concrete advantages of ShareGNN over existing GNN architectures: likely not addressed.
Other minor comments: partially addressed.

Reviewer W22X:
W1-its empirical validation remains modest: likely not addressed.
W2-complexity: likely not addressed.
W3-There is limited exploration of how invariant choices…: likely not addressed.

Reviewer LeQ3:
W1-The method is not scalable: likely not addressed.
W2-The core mechanism appears to be… : likely not addressed.
W3-The main theoretical results… : likely not addressed.

Reviewer NgwU:
W1-The expressive ability, scalability, and transferability are limited: likely not addressed.
W2-High computation cost: likely not addressed.
W3-The performance on real-world datasets is limited: likely not addressed.

**Reviewer Scores:**

For Reviewer fZtg, the initial rating is 4, and it is likely to stay at 4.

For Reviewer W22x, the initial rating is 4, and it is likely to stay at 4.

For Reviewer NgwU, the initial rating is 4, and it is likely to stay at 4.

For Reviewer LeQ3, the initial rating is 4, and it is likely to stay at 4.

---

### Decision · Program_Chairs · 2026-01-26

Reject